# Structural characterization of NrnC identifies unifying features of dinucleases

**Justin D Lormand[1], Soo-Kyoung Kim[2], George A Walters-Marrah[1], Bryce A Brownfield[3], J Christopher Fromme[3], Wade C Winkler[2], Jonathan R Goodson[2], Vincent T Lee[2], Holger Sondermann[1,4,5]\***

[1]Department of Molecular Medicine, Cornell University, Ithaca, United States; [2]Department of Cell Biology and Molecular Genetics, University of Maryland, College Park, United States; [3]Department of Molecular Biology and Genetics, Cornell University, Ithaca, United States; [4]CSSB Centre for Structural Systems Biology, Deutsches Elektronen-Synchrotron DESY, Hamburg, Germany; [5]Christian-Albrechts-Universität, Kiel, Germany

**Abstract** RNA degradation is fundamental for cellular homeostasis. The process is carried out by various classes of endolytic and exolytic enzymes that together degrade an RNA polymer to mono-ribonucleotides. Within the exoribonucleases, nano-RNases play a unique role as they act on the smallest breakdown products and hence catalyze the final steps in the process. We recently showed that oligoribonuclease (Orn) acts as a dedicated diribonuclease, defining the ultimate step in RNA degradation that is crucial for cellular fitness (Kim et al., 2019). Whether such a specific activity exists in organisms that lack Orn-type exoribonucleases remained unclear. Through quantitative structure-function analyses, we show here that NrnC-type RNases share this narrow substrate length preference with Orn. Although NrnC and Orn employ similar structural features that distinguish these two classes of dinucleases from other exonucleases, the key determinants for dinuclease activity are realized through distinct structural scaffolds. The structures, together with comparative genomic analyses of the phylogeny of DEDD-type exoribonucleases, indicate convergent evolution as the mechanism of how dinuclease activity emerged repeatedly in various organisms. The evolutionary pressure to maintain dinuclease activity further underlines the important role these analogous proteins play for cell growth.

**\*For correspondence:**
holger.sondermann@cssb-hamburg.de

**Competing interest:** The authors declare that no competing interests exist.

## Introduction

Traditionally, nano-RNases – enzymes that act on short, typically 2–7 residue-long RNA substrates – have occupied a distinct role in RNA metabolism as they catalyze the final steps in RNA degradation. To date, three main enzyme families have been assigned this function (*Liao et al., 2018*): oligoribonucleases (Orn; *Cohen et al., 2015*; *Orr et al., 2015*; *Zhang et al., 1998*), nano-RNase A and B (NrnA and NrnB; *Fang et al., 2009*; *Mechold et al., 2007*), and nano-RNase C (NrnC; *Liu et al., 2012*). NrnA comprises DHH-DHHA1 domains and is suggested to act as a bidirectional exonuclease that cleaves short RNA fragments from 3′ to 5′ and longer substrates from 5′ to 3′ (*Mechold et al., 2007*; *Schmier et al., 2017*). Orn and NrnC are 3′–5′ exonucleases with a DnaQ fold containing the catalytic DEDD motif (*Chin et al., 2006*; *Yuan et al., 2018*), a domain that is common in enzymes that act on nucleic acids. Notably, Orn (and its eukaryotic ortholog REXO2) or NrnC activity is critical for cellular growth, rendering them unique amongst the exoribonucleases known to date (*Ghosh and Deutscher, 1999*; *Liu et al., 2012*; *Nicholls et al., 2019*).

Despite its classification as a nano-RNase, we recently reported that Orn acts primarily as a diribonuclease, assigning it a highly specific and unique function in clearing the diribonucleotide pool as the

terminal step in RNA degradation. Substrate-bound structures of bacterial and human orthologs, Orn and REXO2, respectively, revealed the scissile bond of the dinucleotide surrounded by the conserved, catalytic DEDDh motif that is involved in divalent cation coordination and catalysis. More importantly, key determinants for the RNA length preference of Orn and REXO2 include a leucine residue that wedges between the two bases of the diribonucleotide and a phosphate cap – invariable residues that coordinate the 5′-phosphate on the substrate and limit substrate length to 2 nucleotides (*Kim et al., 2019*). An independent study on REXO2 confirmed our structural analysis and established the human enzyme as a diribonuclease in mitochondria, where its activity alters gene expression (*Nicholls et al., 2019*), a function that relates to the role of diribonucleotides in transcription initiation (*Druzhinin et al., 2015*; *Goldman et al., 2011*; *Nicholls et al., 2019*; *Vvedenskaya et al., 2012*).

Our previous work on Orn-type RNases was motivated by three main considerations: Orn's essential role for growth in many bacteria, its role in cyclic dinucleotide signaling, and a lack of understanding how substrate specificity towards short RNA substrates, and in turn lack of activity towards longer RNAs, is achieved (*Ghosh and Deutscher, 1999*; *Kamp et al., 2013*; *Kim et al., 2019*; *Orr et al., 2015*; *Palace et al., 2014*). The latter is a basic question as the unique substrate profile is the defining characteristic of this class of enzymes. We demonstrated that Orn has a much higher preference for diribonucleotides compared to 3–7 residue-long RNAs than anticipated previously. This selectivity is due to an active site that is exquisitely suited for diribonucleotides with a 5′ phosphate. The enzyme's diribonuclease activity is required for normal bacterial growth and clearance of c-di-GMP breakdown products. Notably, a knock-out of *Pseudomonas aeruginosa* Orn can not only be complemented by various Orn orthologs, but also by the other three nano-RNases: NrnA, NrnB, and NrnC (*Orr et al., 2018*). However, it is not clear whether this functional complementary correlates with a narrow substrate specificity for dinucleotides.

A recent structural study of NrnC from the soil bacterium *Agrobacterium tumefaciens* identified a homo-octameric assembly as the enzyme's functional unit (*Yuan et al., 2018*). The unit can be divided into two stacked rings, each composed of four NrnC monomers, which together form a central channel lined with eight active sites. Mutagenesis of key active-site residues confirmed the requirement of the conserved DEDDy motif for catalysis and identified positively charged residues lining the channel, which are also important for function. Activity assays indicated single-stranded RNA and DNA as well as double-stranded DNA as potential substrates of NrnC; however, the structural basis for substrate specificity has not been established.

Here, we ask the fundamental question: What is the substrate specificity of NrnC-type enzymes, an RNase that is essential for the growth of Gram-negative bacteria such as *Caulobacter crescentus*, *Bartonella henselae*, and *Brucella abortus* (*Christen et al., 2011*; *Liu et al., 2012*; *Sternon et al., 2018*). We present crystal structures of NrnC from *B. henselae* and *Brucella melitensis* in their substrate-bound and apo states. The structures confirm an octameric assembly as a conserved feature of NrnC-type RNases. The substrate-bound states reveal, similar to Orn, a narrow active site that appears optimized for dinucleotides. This preference is reflected also in the enzyme's activity profile. A comparative genomics analysis indicates that Orn and NrnC, despite using a common DnaQ fold, evolved separately as isofunctional enzymes (*Galperin and Koonin, 2012*; *Omelchenko et al., 2010*). Considering also the distribution of the structurally unrelated, yet functionally overlapping NrnA- and NrnB-type RNases predominantly in organisms that lack Orn and NrnC underlines the importance to maintain diribonuclease activity for cellular function in the bacterial and eukaryotic domains of life.

## Results and discussion
### Overall structure of diribonucleotide-bound NrnC

The earlier observation that NrnC expression is able to complement a *P. aeruginosa orn* deletion strain indicates that both enzymes function on diribonucleotides (*Orr et al., 2018*), consistent with their initial classification as nano-RNases (*Datta and Niyogi, 1975*; *Liu et al., 2012*; *Niyogi and Datta, 1975*; *Yu and Deutscher, 1995*). However, while we showed that Orn acts as a dedicated diribonuclease (*Kim et al., 2019*), the structural basis for NrnC's substrate preference remained not well defined. Specifically, it was not clear how NrnC distinguishes between short RNAs and longer polymers. To answer this question, we determined crystal structures of wild-type *B. henselae* and *B. melitensis* NrnC (NrnC$_{Bh}$ and NrnC$_{Bm}$, respectively) bound to pGG and, in the case of NrnC$_{Bm}$, also

in the substrate-free state. $NrnC_{Bh}$ forms a homo-octameric assembly comprising two C4-symmetric rings (*Figure 1A*) as observed with the previously determined substrate-free, orthologous protein from *A. tumefaciens* (69% sequence identity compared to $NrnC_{Bh}$, monomer/octamer all-atom rmsd 0.4/1.3 Å; PDB:5ZO3; *Yuan et al., 2018*). The two rings stack with the same face, tail-to-tail, forming a D4-symmetric octamer. The contacts between the subunits within each ring are dominated by a few polar interactions spanning between 441 and 512 Å$^2$ (*Figure 1—figure supplement 1A and B*; determined by PISA [*Krissinel and Henrick, 2007*]). In contrast, pairwise, homotypic interactions between the two rings involve an extensive hydrophobic interface of 1204 Å$^2$ via an antiparallel packing of the last helix of the NrnC fold. This mode of ring stacking positions the C-terminus of one monomer so that it reaches into the active site of the adjacent monomer in the other ring (*Figure 1—figure supplement 1A and C*).

The diribonucleotide pGG is bound to all eight active sites of the $NrnC_{Bh}$ octameric assembly (*Figure 1B*). The active sites face the center of the central cavity formed by the NrnC octamer, positioned mid-way of each ring. In this crystallographic state, the residues of the catalytic DEDDy signature motif ($D^{25}$, $E^{27}$, $D^{84}$, $D^{155}$, $Y^{151}$) are primed for accepting divalent metal ions for nucleotide hydrolysis. The side chain of $Y^{151}$ coordinates a water molecule, which likely serves as the attacking nucleophile in the reaction (*Figure 1C*).

The structure also reveals the molecular basis for substrate coordination. Reminiscent of Orn's active site, the bases of the diribonucleotide are splayed apart by a leucine residue, or L-wedge ($L^{31}$) (*Figure 1C*). Continuing with the parallels to Orn, the 5′ phosphate of pGG is coordinated by several residues forming a 'phosphate cap,' in this case basic residues $H^{79}$, $K^{103}$, and $H^{205}$, the latter being the second to last residue of the protein, contributed from a subunit from the adjacent ring. The specific motifs coordinating the substrate are invariable in NrnC orthologs, in contrast to the exterior surface with overall lower conservation (*Figure 1D*), and culminate in a length-restricted active site that appears optimized for diribonucleotides. A comparison with the structurally related Rrp6 exonuclease subunit of the exosome bound to a longer RNA substrate supports this notion as residues $K^{103}$ and $H^{205}$ of NrnC's phosphate cap directly block the path for longer substrates (*Figure 1—figure supplement 2A and C*; *Wasmuth et al., 2014*). Similarly, RNase D, a homologous exoribonuclease that processes longer and stable RNA molecules, presents a more expansive, open active site, although the lack of a substrate-bound structure prevents a more direct comparison of this state (*Figure 1—figure supplement 2B and C*; *Zuo et al., 2005*).

Similar binding poses to pGG at NrnC were observed with pAA and pGC (*Figure 1—figure supplement 3A–C*), suggesting that most if not all diribonucleotides can be accommodated by NrnC. A co-crystal structure with the di-phosphorylated mononucleotide pAp, a metabolite described as an inhibitor of NrnC (*Liu et al., 2012*), shows the ligand predominantly occupying the 5′ position of the active site with the 3′ phosphate engaging the catalytic motif and the 5′ phosphate being coordinated by the phosphate cap of NrnC (*Figure 1—figure supplement 3D*).

To establish the functional relevance of the crystallized state of NrnC, we initiated *in crystallo* catalysis by soaking $NrnC_{Bh}$•pGG co-crystals in solutions with divalent cations, magnesium ($Mg^{2+}$) or manganese ($Mn^{2+}$). The enzyme became catalytically active with the addition of either cation, with the resulting electron densities showing a broken phosphodiester bond (*Figure 1—figure supplement 4*). The experiments also confirmed a two-metal mechanism first by interpretation of electron density upon $Mg^{2+}$ soaking (*Figure 1—figure supplement 4A*). The placement of active site metal atoms was subsequently confirmed using anomalous data collected on $Mn^{2+}$-soaked crystals (see anomalous difference map, *Figure 1—figure supplement 4B*). In these post-catalysis structures, the 5′ GMP appears to leave the active site first, as suggested by weaker electron density indicative of lower mononucleotide occupancy at the 5′ site compared to the 3′ site.

## The characteristic active-site motifs of $NrnC_{Bh}$ contribute to diribonuclease activity

The structural analysis revealed the molecular basis for substrate binding to NrnC, identifying molecular features that constrain the active site. To assess their relevance for NrnC's catalytic activity, we tested tag-less, purified $NrnC_{Bh}$ and structure-based point mutants thereof in an in vitro activity assay. All mutant proteins retained their quaternary structure and purified as octamers, indistinguishable from wild-type NrnC (*Figure 2—figure supplement 1*). 5′-$^{32}$P-radiolabeled pGG was incubated with

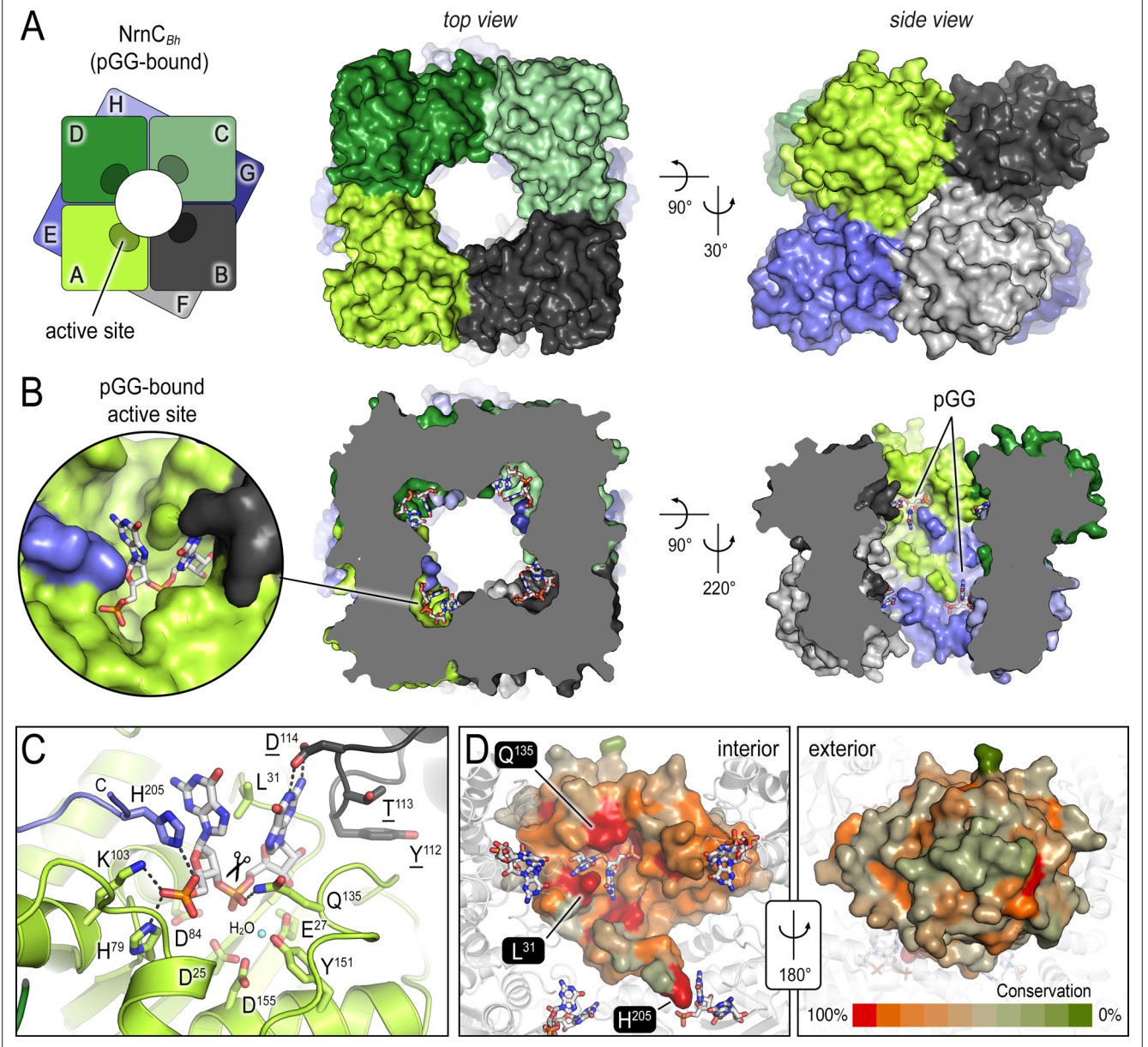

**Figure 1.** The crystal structures of *B. henselae* nano-RNase C (NrnC) bound to pGG reveals motifs defining substrate specificity. (**A**) The octameric assembly. NrnC_{Bh} is shown as surface representation in two views. Each monomer is shown in a distinct color. The cartoon illustrates the stacking of the two tetrameric NrnC rings that form the octamer with a central, round opening. (**B**) Active-site position. Each monomer contributes one active site, here bound to the substrate pGG, facing toward NrnC's central pore. Each active site includes a C-terminal tail of a subunit from an adjacent ring. (**C**) Substrate coordination. The catalytic DEDDy motif and residues coordinating each moiety of pGG contacts are shown as sticks, with carbon residues colored according to monomer identity. Residue Y^{151} coordinates water molecule near the scissile bond. (**D**) Conservation mapping on a surface representation of a NrnC monomer. Conservation scores were calculated based on a multisequence alignments (MUSCLE; *Edgar, 2004*) of NrnC homologs identified using a sequence search on the EggNOG resource, version 5.0.0 (*Huerta-Cepas et al., 2019*) and the sequence of NrnC_{Bh} as the input. Outliers were identified based on sequence length and non-consensus insertions, resulting in a final collection of 560 sequences of putative NrnC orthologs. The two views, separated by a 180° rotation, show the cavity-facing (interior, left) and outer-facing (exterior, right) surface regions.

The online version of this article includes the following figure supplement(s) for figure 1:

**Figure supplement 1.** Inter- and intra-ring contacts in the NrnC_{Bh} octamer.

**Figure supplement 2.** Comparison of nano-RNase C (NrnC) to structurally related proteins reveals the constricted nature of NrnC's active site.

**Figure supplement 3.** Structural comparison of NrnC_{Bh} bound to various ribonucleotides.

**Figure supplement 4.** *In crystallo* catalysis indicates a two-metal mechanism of NrnC_{Bh} activity.

wild-type or mutant enzymes in the presence of divalent cations at physiological ionic strength. Quenched reactions were resolved via urea-denaturing PAGE to observe nucleolytic cleavage over time (*Figure 2A*). With wild-type NrnC$_{Bh}$, the majority of pGG was processed already by the first time point at 30 s, and complete cleavage of pGG to GMP was achieved by 3 min. Nucleolytic activity on pGG was completely inhibited by alanine mutation of the catalytic DEDDy motif residues D$^{25}$ and Y$^{151}$. Intermediate cleavage kinetics were evident with proteins with disrupted leucine wedge (NrnC$_{Bh}$-L$^{31}$A) as well as the phosphate cap (NrnC$_{Bh}$-H$^{79}$A, -K$^{103}$A, or -H$^{205}$A). NrnC binding to and activity on pGG were slightly inhibited by pAp (*Figure 2B and C*). In contrast, the diguanylate compound GG, which lacks a 5′ phosphate, did not inhibit NrnC's binding to or activity on pGG (*Figure 2B and C*), further elaborating on the importance of the interaction between phosphate cap residues and the 5′ phosphate for NrnC function.

We previously proposed a model of cellular fitness in which the loss of *orn* leads to toxic diribonucleotide accumulation that is detrimental to the cell. The *orn* deletion in *P. aeruginosa* manifests as small-colony growth, which is reversible by complementation with *orn* or other nano-RNases expressed from a plasmid (*Figure 2D*; *Kim et al., 2019*; *Orr et al., 2018*). Here we used the rescue of the small colony phenotype as a readout of NrnC diribonuclease activity in cells, quantified as colony size. Complementation of the deletion strain with wild-type NrnC$_{Bh}$ (with a C-terminal HA tag for detection) restored normal colony size, while NrnC alleles containing mutations within the DEDDy motif failed to complement (*Figure 2D*). Further, NrnC alleles containing mutations in the L-wedge or the phosphate cap showed reduced complementation effects (L$^{31}$A, K$^{103}$A, K$^{132}$A, or H$^{205}$A; *Figure 2D*). NrnC$_{Bh}$-H$^{79}$A failed to complement the *orn* deletion. Western blot analysis established protein expression for all mutant and wild-type NrnC variants, with the exception of NrnC$_{Bh}$-H$^{79}$A, which expressed poorly in *P. aeruginosa*, preventing a distinction between failure to rescue because of the mutation or protein levels, or both (*Figure 2—figure supplement 2*). Together, these data confirm the importance of the motifs identified in the substrate-bound NrnC structures for the enzyme's diribonuclease activity in vitro and in cells.

## Structural comparison of NrnC substrate-bound states reveals an active site optimized for dinucleotides

To further understand the structural basis of NrnC's substrate preference, we determined structures of *B. melitensis* NrnC (NrnC$_{Bm}$) in several apo and partially pGG-bound states. The overall structure of pGG-bound NrnC$_{Bm}$ is virtually identical to NrnC$_{Bh}$, namely a homo-octameric assembly with eight active sites pointing toward the central channel (*Figure 3A and B*). The acidic active site residues (D$^{24}$, E$^{26}$, D$^{83}$, D$^{154}$, and Y$^{150}$) as well as the L-wedge (L$^{30}$) and phosphate cap (H$^{78}$, K$^{102}$, and H$^{204}$) are structurally and functionally conserved.

NrnC$_{Bm}$ incubated with pGG crystallized with two molecules per asymmetric unit, only one of which was bound to the dinucleotide. The resulting octameric assembly contains alternating apo- and pGG-bound subunits per tetrameric ring (*Figure 3B*). This mixed-state structure allowed us to propose features modulating substrate binding. In substrate-bound monomers (including in the structures of NrnC$_{Bh}$), the DEDDy residue Y$^{150}$ points inward toward the scissile phosphodiester bond, coordinating the attacking water. In contrast, the Y$^{150}$ side chain points away from the active site in substrate-free NrnC (*Figure 3A and B*). A loop from an adjacent subunit that mediates inter-ring contacts between the monomers and that includes residue D$^{113}$ buttresses the 3′ base of the substrate. While the octameric assembly remains in the absence of substrate, this loop moves outward from the active site and D$^{113}$ rotates away from the substrate (*Figure 3D*). The most drastic conformational change however is attributed to a flexible loop spanning residues $^{130}$SKQQQS$^{135}$. This loop is ordered and positioned in contact with the dinucleotide in both ortholog structures (*Figures 1C and 3B and C*). In this state, residue Q$^{134}$ of NrnC$_{Bm}$ (or Q$^{135}$ in NrnC$_{Bh}$) contacts the scissile phosphate via a hydrogen bond; K$^{131}$ (or K$^{132}$ in NrnC$_{Bh}$) points toward the 5′ phosphate, thus contributing to the phosphate cap. In contrast, in the substrate-free state NrnC$_{Bm}$, captured in the mixed-state structure or a homogeneous apo-state structure, this loop swings away from the active site or is completely disordered, leading to an overall widening of the active site (*Figure 3A*; *Figure 1—figure supplement 2C*).

In a second apo-NrnC$_{Bm}$ crystal form, we captured an alternate conformation in which the conserved hydrophobic residue F$^{79}$ moves into the active site (*Figure 3—figure supplement 1*). The movement of this residue, which is adjacent to the phosphate cap residue H$^{78}$, is realized through a flip of the

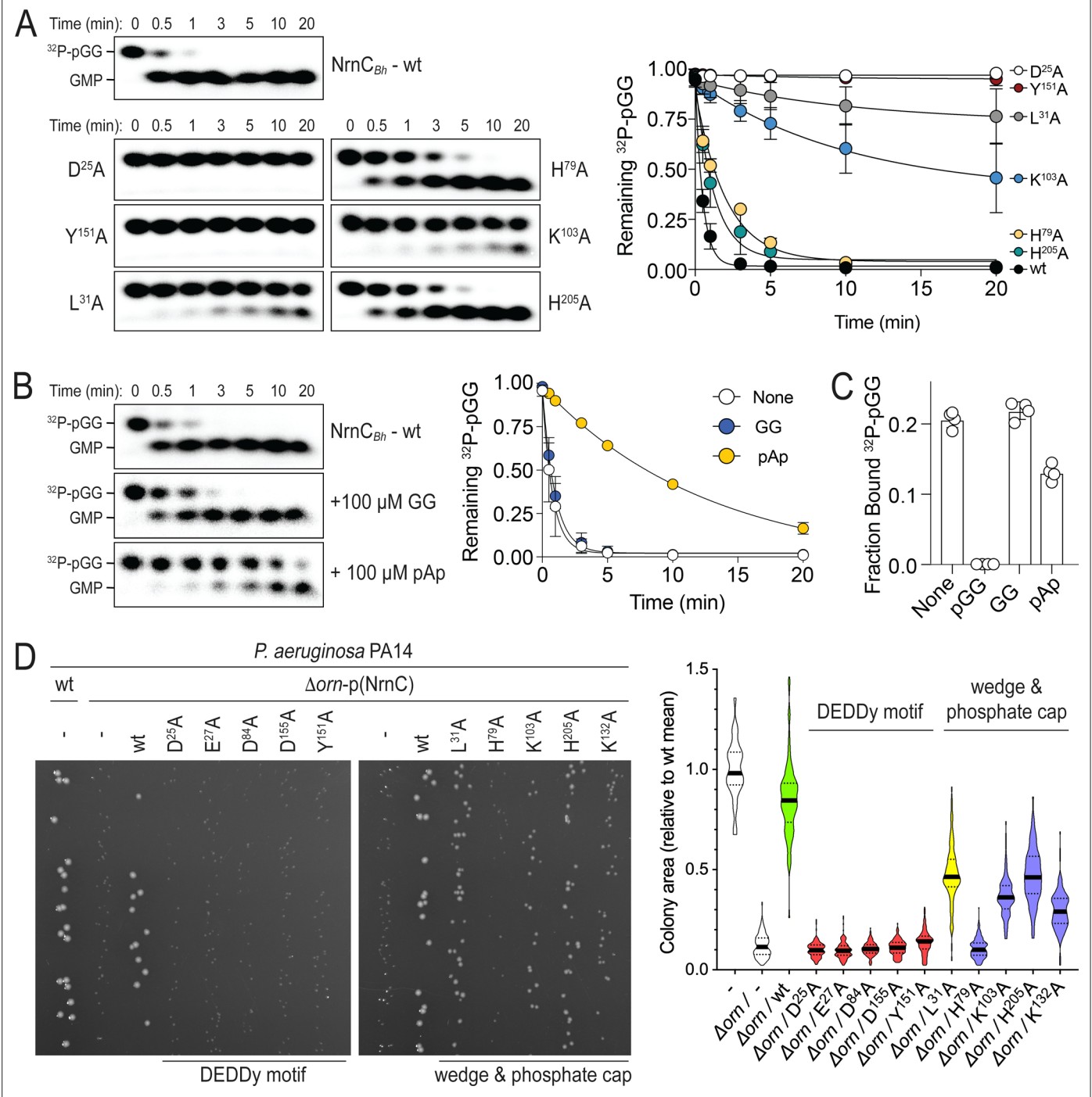

**Figure 2.** Phosphate cap and L-wedge contribute to nano-RNase C's (NrnC's) diribonuclease activity. (**A**) In vitro enzyme activity. Degradation of $^{32}$P-pGG (1 μM total) by purified wild-type NrnC$_{Bh}$ or variants with alanine substitutions (5 nM) at the indicated sites was assessed. Samples were stopped at the indicated times (min) and analyzed by denaturing 20% PAGE. Representative gels are shown (left). The graph (right) shows the means and SD of three independent experiments. (**B**) Effect of a dinucleotide lacking the 5′ phosphate (GG) and pAp on NrnC catalysis. pGG processing was assessed as in (**A**) but in the presence or absence of 100-fold excess (over $^{32}$P-pGG) GpG or pAp. Representative gels (left) and quantification from three independent experiments (right) are shown. Means and SD are plotted. (**C**) Competition binding studies. Fraction bound of $^{32}$P-pGpG to 200 nM purified NrnC$_{Bh}$ in the presence of no competitor, 100 μM pGG, 100 μM GpG, or 100 μM pAp is plotted as individual data, means, and SD of four independent experiments. (**D**) Complementation of the small-colony phenotype of *P. aeruginosa* Δ*orn* by wild-type and mutant NrnC$_{Bh}$. Bacterial cultures were diluted and dripped on LB agar plates. After overnight incubation, representative images of the plates were taken (left). Experiments were performed in triplicate. Quantification of respective colony sizes is shown as violin plots (right).

*Figure 2 continued on next page*

*Figure 2 continued*

The online version of this article includes the following figure supplement(s) for figure 2:

**Source data 1.** Source data for *Figure 2A*.

**Source data 2.** Source data for *Figure 2B*.

**Source data 3.** Source data for *Figure 2C*.

**Source data 4.** Source data for *Figure 2D*.

**Figure supplement 1.** SEC-MALS of NrnC$_{Bh}$ wild-type and mutant variants.

**Figure supplement 2.** Expression of NrnC$_{Bh}$ wild-type and mutant variants in *P. aeruginosa Δorn*.

**Figure supplement 2—source data 1.** Original and labeled, unedited western blot image.

peptide backbone and would introduce a clash with the nucleotide. In the same structure, the flexible SKQQQS loop is collapsed into the active site, trapping the catalytic residue Y$^{150}$ in an intermediate conformation, which would introduce further clashes with the nucleotide substrate. Simultaneously, a rotamer change of the phosphate-cap residue H$^{204}$ pivots its sidechain away from the active site, opening it for access to substrates (*Figure 3—figure supplement 1*). This conformation may depict a post-hydrolysis state, suggesting a mechanism for product release.

## Analysis by cryo-electron microscopy supports a narrow substrate preference of NrnC in solution

Efforts to crystallize NrnC bound to longer substrates yielded structures with either empty active sites or with only pGG being resolved in the resulting electron densities. Residual RNase activity over the course of the crystallization or an impact of the longer substrates on crystal packing could contribute to the inability to resolve longer substrates, assuming they bind in the first place. As an alternative to crystallography, we determined NrnC$_{Bh}$ structures bound to pGG, pAGG, and pAAAGG by cryo-electron microscopy (cryo-EM), a technique that can visualize complexes formed after short equilibration periods and does not rely on proteins packing in a lattice. Considering that NrnC forms an octamer as the biological unit, we processed each data set with C1 and D4 symmetry, with the resulting models consisting of eight independent or an averaged chain, respectively (resolutions range from 2.72 to 3.27 Å; *Figure 4*, *Figure 4—figure supplements 1–8*). Processing with C1 symmetry preserves the individuality of each monomer allowing the observation of differences, for example, between active sites within the octamer. Applying D4 symmetry during processing averages all eight monomers, yielding a consensus model; however, regions with conformational differences between individual monomers may contribute to apparent disorder in the electron density maps. In an additional experiment with the 5-mer pAAAGG as the substrate, CaCl$_2$ was added to the buffer to probe whether the addition of Ca$^{2+}$ ions, which prevent catalysis but can support substrate binding, has an effect on the binding of longer RNAs.

The structure of pGG-bound NrnC$_{Bh}$ confirmed all active site features described above based on the crystallographic data, namely a narrow active site, a phosphate cap coordinating the 5′ phosphate of the substrate, a L-wedge splaying apart the two bases, and an active-site facing, well-resolved SKQQQS loop (*Figure 4A*, *Figure 4—figure supplement 9A*). Refinement with lower and higher symmetry resulted in comparable density maps, indicating a consensus state with eight nearly identical active sites.

The density maps of NrnC bound to any of the longer substrates resolved invariably only a diribonucleotide at the active site (with poorer density indicating the position of the ribose of the third nucleotide from the 3′ end) (*Figure 4B*). The remainder of the longer substrates appeared disordered. Using D4-symmetry-averaged data, we also noticed consistently disorder of the SKQQQS loop (*Figure 4B*). Inspection of symmetry-less (C1) density maps revealed different loop conformations in the individual monomers of the octamers that, when symmetry-averaged, result in the apparent disorder (*Figure 4—figure supplements 5–8*). Notably, the majority of monomers contain a disengaged loop conformation, leaning away from the active site. Addition of Ca$^{2+}$ with the pAAAGG substrate results in increased ordering of the loop in a disengaged state, similar to that observed in the substrate-free crystal structures of NrnC$_{Bm}$ (*Figure 3E*, *Figure 4—figure supplement 9B*). Together, these results suggest that longer substrates may bind NrnC, but only the first two 3′ nucleotides are

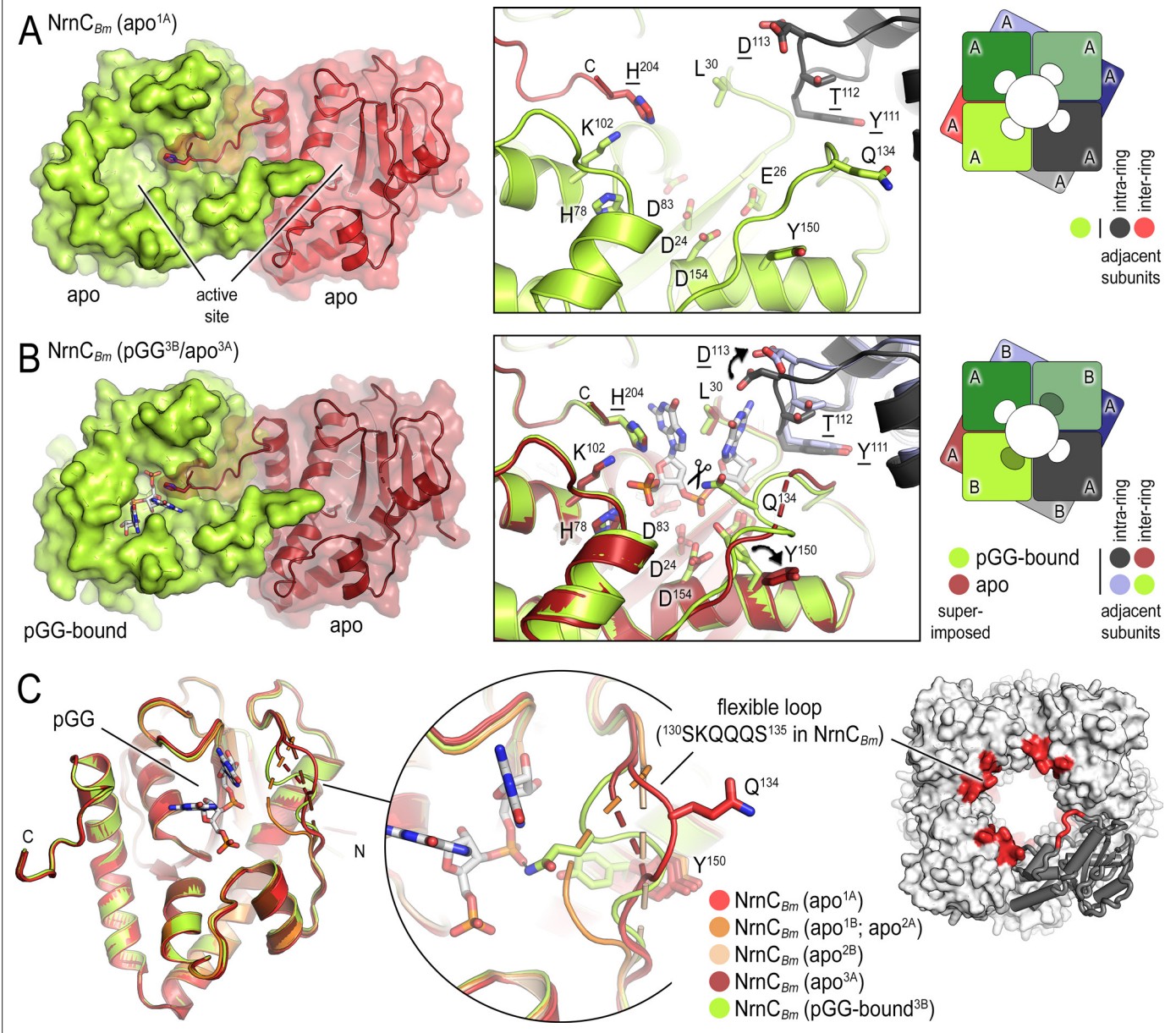

**Figure 3.** *B. melitensis* nano-RNase C (NrnC) crystal structures reveal a flexible loop that constraints the enzyme's active site. (**A**) Crystal structure of apo-NrnC$_{Bm}$. A crystallographic dimer as part of the octameric assembly is shown as surface presentation (left) and close-up of the active site (middle). The diagram (right) depicts the octamer and the spatial relationship of the monomers shown. (**B**) Crystal structure of NrnC$_{Bm}$ with alternating substrate-bound and empty active sites. The close-up (middle) shows a superposition of the two monomers in the asymmetric unit, depicting their conformational difference and adjacent monomers, with intra- and inter-ring neighbors colored as shown in the diagram (right). (**C**) Superposition of four apo-NrnC$_{Bm}$ conformations based on three independent crystal forms (comprising chains apo$^{1A}$/apo$^{1B}$ for form 1; apo$^{2A}$/apo$^{2B}$ for form 2, and apo$^{3A}$/pGG-bound$^{3B}$ for form 3), compared to the pGG-bound conformation of the same protein shown in (**B**). The position of the flexible loop (red) in the NrnC octamer is shown (right panel).

The online version of this article includes the following figure supplement(s) for figure 3:

**Figure supplement 1.** Overlay of an alternative crystallographic apo-NrnC$_{Bm}$ state with the apo- and pGG-bound states observed in the crystal structure shown in *Figure 3B*.

well coordinated at the active site. Furthermore, RNAs with more than two residues in length increase conformational variability at the active site, likely impacting catalytic activity toward those substrates. The SKQQQS loop may act as an activation loop that needs to swing into the active site, adopting a distinct, ordered conformation. The conformational change, induced by dinucleotide binding,

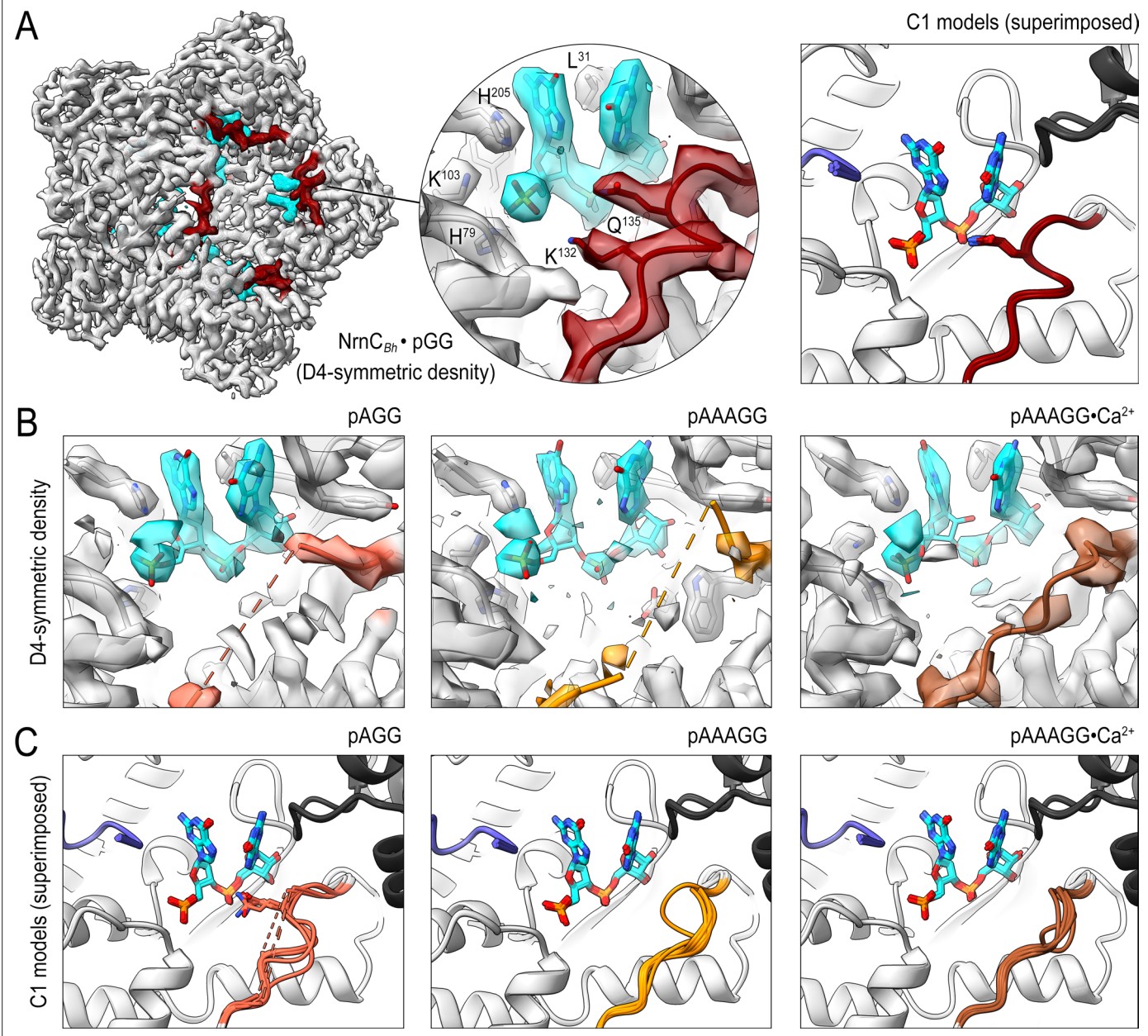

**Figure 4.** Cryo-electron microscopy (cryo-EM) structures of NrnC$_{Bh}$ with 2-, 3-, 5-mer RNA substrates show substrate length-dependent active site conformations. (**A**) Electron density map of a NrnC$_{Bh}$ octamer in complex with pGG. D4 symmetry was applied during final map refinement. pGG molecule and density are colored cyan. The SKQQQS-containing loops (residues 130–137) are colored maroon. Superposition of all eight active sites from a reconstruction with C1 symmetry (right panel) shows consensus order in the loop when bound to pGG. (**B**) Active site images shown for NrnC$_{Bh}$ incubated with 3-mer and 5-mer (with or without Ca$^{2+}$) RNA substrates. Regions corresponding to those shown in (**A**) are shown in color, with light red (left panel), orange (middle panel), and brown (right panel) depicting the loop/loop density from structures determined with added pAGG, pAAAGG, and pAAAGG•Ca$^{2+}$, respectively. D4-symmetric maps are shown. (**C**) Superposition of all eight active sites from octamer reconstructions based on respective C1-symmetric maps for each RNA substrate. Color scheme is as described in (**B**).

The online version of this article includes the following figure supplement(s) for figure 4:

**Figure supplement 1.** Cryo-electron microscopy (cryo-EM) workflow and resolution for NrnC$_{Bh}$•pGG.

**Figure supplement 2.** Cryo-electron microscopy (cryo-EM) workflow and resolution for NrnC$_{Bh}$•pAGG.

**Figure supplement 3.** Cryo-electron microscopy (cryo-EM) workflow and resolution for NrnC$_{Bh}$•pAAAGG.

**Figure supplement 4.** Cryo-electron microscopy (cryo-EM) workflow and resolution for NrnC$_{Bh}$•pAAAGG in the presence of Ca$^{2+}$ ions.

**Figure supplement 5.** Overall and individual active site electron density of a NrnC$_{Bh}$•pGG octamer after refinement with C1 symmetry.

*Figure 4 continued on next page*

*Figure 4 continued*

**Figure supplement 6.** Overall and individual active site electron density of a NrnC$_{Bh}$•pAGG octamer after refinement with C1 symmetry.

**Figure supplement 7.** Overall and individual active site electron density of a NrnC$_{Bh}$•pAAAGG octamer after refinement with C1 symmetry.

**Figure supplement 8.** Overall and individual active site electron density of a NrnC$_{Bh}$•pAAAGG octamer in the presence of Ca$^{2+}$ ions after refinement with C1 symmetry.

**Figure supplement 9.** The conformation of nano-RNase C (NrnC) bound to substrates with more than two bases resembles the crystallographic apo-state.

correlates with the positioning of the catalytic tyrosine residue of the DEDDy motif in a catalytically competent position (also illustrated in *Figure 3—figure supplement 1*). Likewise, the SKQQQS loop would have to move out of the way to allow for substrate binding. In summary, the combined structural data indicate NrnC is optimized for dinucleotide processing over longer substrates, mirroring our analysis of Orn and REXO2 (*Kim et al., 2019*; *Nicholls et al., 2019*).

## NrnC$_{Bh}$ acts preferentially on dinucleotides

Although the structural analysis revealed a narrow active site akin to that of Orn, NrnC's substrate length preference has not been assessed formally in this new context. To address this, we conducted kinetic experiments with RNAs of increasing length as substrates, following protocols established for Orn (*Kim et al., 2019*). In the initial assay, RNAs with two or more residues were in 200-fold abundance over NrnC, a condition where NrnC turned over the entire pGG pool within 1 min (*Figure 5A*). In comparison to NrnC's expedient activity on pGG, increasing the substrate length by only a single residue (pAGG) resulted in a striking decrease of nucleolytic cleavage under otherwise identical conditions. For RNAs with four and more residues, the substrate was processed incompletely and a band indicative of a cleavage product that was one base shorter at the 3′ end slowly increased over the course of the experiment (*Figure 5A and B*). Increasing the concentration of NrnC increased activity on the longest substrate tested, an RNA with seven bases (5′-$^{32}$P-labeled AAAAAGG, pAAAAAGG), but full conversion to mononucleotides required a ratio of 1:1 NrnC:RNA, indicating a comparatively inefficient, and likely less physiological mechanism (*Figure 5B*). Furthermore, and similar to the kinetics observed with Orn (*Kim et al., 2019*), a diribonucleotide intermediate was undetectable with enzyme concentrations that were required to observe the successive breakdown of the longer RNA, revealing the rapid turnover of dinucleotides, proceeding at much faster timescales than with any other intermediate that could be readily observed (*Figure 5B*).

Quantification of NrnC's binding to radiolabeled RNA substrates of different length agreed with the preference of NrnC to cleave diribonucleotides. Affinities of NrnC for radiolabeled substrates (2–7-mer RNA) were determined at physiological ionic strength and in the presence of Ca$^{2+}$ to inhibit any potential residual catalysis (*Rosta et al., 2014*). NrnC$_{Bh}$ bound to pGG with a $K_d$ = 17.7 nM. Increasing substrate length by just one additional residue resulted in a nearly 200-fold decrease in affinity ($K_d$ pAGG = 3.49 μM; *Figure 5C*). RNA substrates of four, five, or six residues showed similar decreases in affinity, while a 7-mer RNA substrate showed intermediate binding strength with a 32-fold decrease from pGG ($K_d$ pAAAAAGG = 576 nM). As another method to assess substrate preference, competition experiments were performed by incubating NrnC$_{Bh}$ with $^{32}$P-pGG with or without unlabeled RNAs of different length. While unlabeled pGG was able to displace radiolabeled pGG quantitatively on NrnC, longer substrates were less potent competitors under otherwise identical conditions (*Figure 5—figure supplement 1*). Together, the combined structural and biochemical results argue for a strong preference of NrnC toward the shortest species of RNAs, diribonucleotides with a 5′ phosphate.

## NrnC$_{Bh}$ processes DNA under specific experimental conditions

The *A. tumefaciens* NrnC octamer was previously interpreted as a conduit for long, polymeric substrates, in particular single-stranded RNA and single-stranded or double-stranded DNA (dsDNA), based on the octamer's channel dimensions and positioning of the active sites (*Yuan et al., 2018*). DNase activity was proposed to allow NrnC octamers to act on opposite ends of dsDNA to completely unwind and degrade it by passing the strand through the central channel. Here, we asked whether NrnC$_{Bh}$ could act on DNA substrates (*Figure 5—figure supplement 2*). Under near-physiological ionic

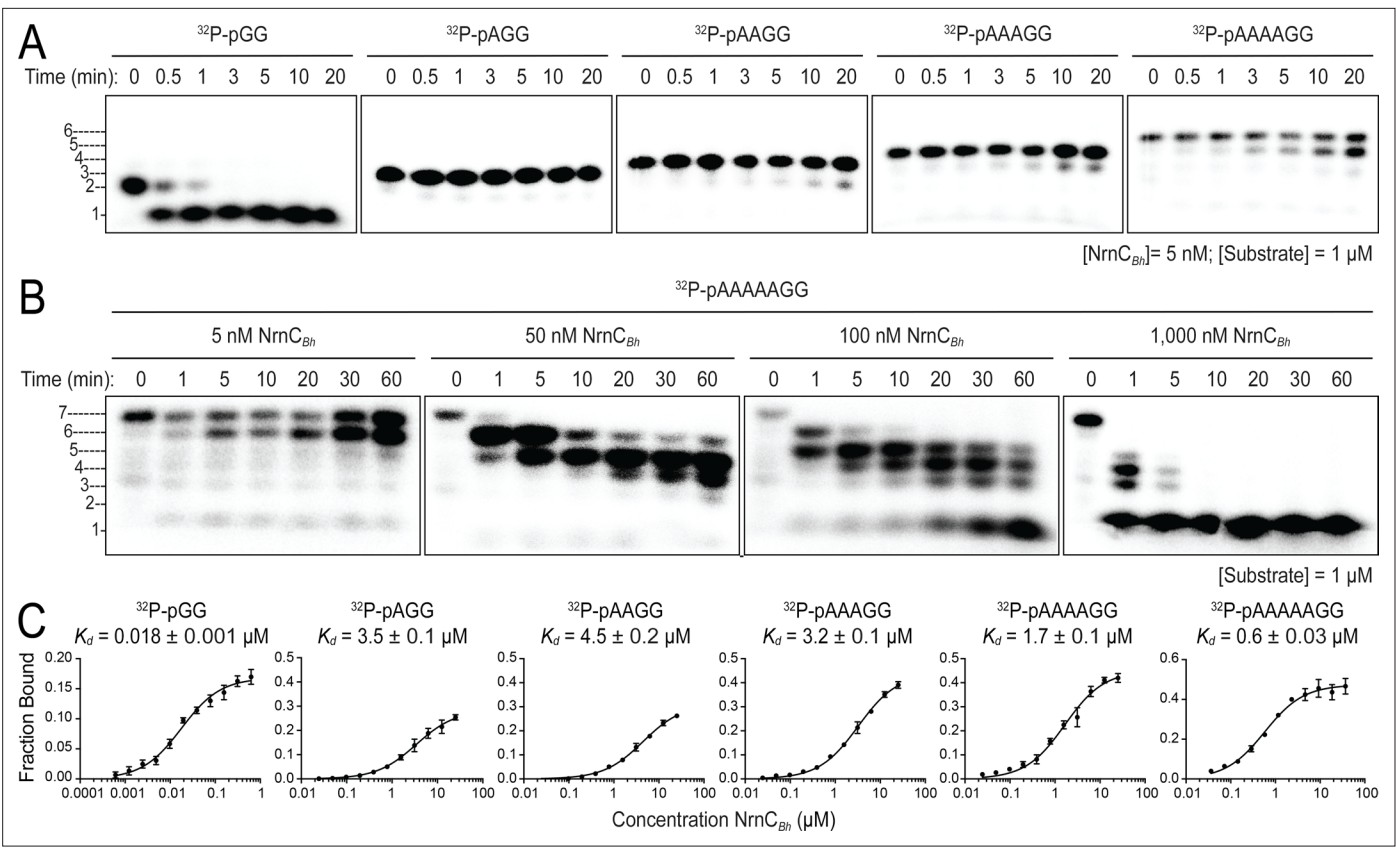

**Figure 5.** Nano-RNase C (NrnC) shows a strong preference for substrates with two residues in length. (**A**, **B**). RNase assays. Experiments are similar to those in *Figure 2* but were performed with radiolabeled substrates from 2 to 7 residues in length. Representative gels of at least two independent experiments are shown. In (**B**), enzyme concentration was varied from 5 to 1000 nM (1:200 to 1:1 enzyme:substrate ratio). Substrate length-dependent binding studies. (**C**) Affinity of NrnC for RNA with different lengths. Fraction bound of radiolabeled substrates of increasing length was assessed at different NrnC concentrations and is plotted as means and SD from three independent experiments.

The online version of this article includes the following figure supplement(s) for figure 5:

**Source data 1.** Source data for *Figure 5A*.

**Source data 2.** Source data for *Figure 5B*.

**Source data 3.** Source data for *Figure 5C*.

**Figure supplement 1.** Competition binding studies.

**Figure supplement 1—source data 1.** Quantification of $^{32}$P-pGG binding to nano-RNase C (NrnC) in the presence of unlabeled RNA with increasing length (in three replicates).

**Figure supplement 2.** NrnC$_{Bh}$ degrades long DNA fragments under distinct conditions.

**Figure supplement 2—source data 1.** Original, unedited agarose gel images and composite overview of nano-RNase C (NrnC) activity against 1.5 kb, double-stranded DNA substrates.

**Figure supplement 3.** The preferred substrates of NrnC$_{Bh}$ are diribonucleotides and deoxy-dinucleotides.

**Figure supplement 3—source data 1.** Source data for *Figure 5—figure supplement 3B*.

**Figure supplement 3—source data 2.** Source data for *Figure 5—figure supplement 3C*.

**Figure supplement 3—source data 3.** Source data for *Figure 5—figure supplement 3D and E*.

strength and in the presence of either Mg$^{2+}$ or Mn$^{2+}$ ions, NrnC$_{Bh}$ failed to degrade a 1.5-kb-long dsDNA fragment (*Figure 5—figure supplement 2B and C*). Degradation of dsDNA was only observed at low ionic strength (~0–60 mM NaCl) and only in the presence of Mn$^{2+}$ ions, conditions that match those used for *A. tumefaciens* NrnC. The requirement for Mn$^{2+}$ for activity on DNA substrates mirrors the previously reported phenomenon observed with the 3′–5′ exonuclease EXD2 that acts on both ribo- and deoxyribonucleotides (*Park et al., 2019*).

To examine whether potential NrnC activity on dsDNA was dependent on the presence of a 5′ phosphate or a 5′ or 3′ overhang, dsDNA fragments were treated either with T4 polynucleotide kinase (PNK), calf intestinal alkaline phosphatase (CIP), or restriction enzymes (NdeI, NotI, or KpnI) as indicated (*Figure 5—figure supplement 2A and D*). With any of these modifications and similar to the parent, blunt dsDNA, degradation was only observed in combination of the absence of NaCl and presence of $Mn^{2+}$ ions. To rule out double-stranded substrate length dependence, we assessed whether $NrnC_{Bh}$ is capable of degrading annealed 27-nucleotide-long dsDNA or dsRNA with either a 5′ or 3′ overhang. Under conditions at which $NrnC_{Bh}$ cleaves pGG within a minute (*Figure 5A*), no activity against double-stranded oligonucleotides was observed (*Figure 5—figure supplement 2* and *Figure 5—figure supplement 3*). Blunt plasmid DNA and 27 bp DNA with a 5′ or 3′ overhang also failed to affect pGG degradation by $NrnC_{Bh}$ to an appreciable level, indicating that dsDNA does not act as an inhibitor of NrnC activity (*Figure 5—figure supplement 3D*). Finally, we compared the enzyme's activity against diribonucleotides pGG and pAA to that against deoxy-dinucleotide pdAdA or a single-stranded DNA trinucleotide, pdAdGdG. $NrnC_{Bh}$ phosphodiesterase activity on diribonucleotides was sequence independent, with both pGG and pAA being processed following similar kinetics (*Figure 5—figure supplement 3E*). This result correlates with the structural analysis that revealed similar binding poses of these substrates at NrnC's active site (*Figure 1—figure supplement 3*). NrnC also cleaved pdAdA, nano-DNA with 2 nucleotides in length, at the same rate as diribonucleotides of the same sequence (*Figure 5—figure supplement 3E*). As was observed with RNA substrates, single-stranded DNA with more than two nucleotides were appreciably poorer substrates (*Figure 5—figure supplement 3E*).

Together, these experiments call into question a general DNase activity of NrnC, especially against dsDNA, although such an activity under specific cellular conditions remains plausible. Conversely, the studies indicate NrnC as a general dinuclease that cleaves deoxy-dinucleotides or diribonucleotides, a distinction to Orn and Rexo2 that act preferentially on diribonucleotides (*Nguyen et al., 2000*; *Mechold et al., 2006*; *Chu et al., 2019*). The functional relevance of an octameric NrnC assembly remains not fully understood. From the structural analysis, lateral subunit interactions within the octamer block off the active site at the 3′ end of the substrate (*Figure 1*), consistent with 3′–5′ exonuclease activity of the enzyme. But it is also possible that the octamer creates a nano-compartment for the efficient attraction and degradation of the smallest nano-RNase fragments, dinucleotides, with the central channel acting as a steric constriction for longer substrates.

## Phylogenetic analysis indicates repeated evolution of critical dinuclease activity

After having elucidated the mechanisms that signify NrnC-type and Orn-type RNases as dedicated dinucleases, we reinvestigated the evolutionary distribution of these enzymes, relative to their structural homologs RNase D and RNase T, respectively. We included NrnA and NrnB in this analysis as the only other general RNases unrelated to NrnC and Orn, which also could stand in for Orn in a *P. aeruginosa* deletion strain, as shown previously (*Orr et al., 2018*). We identified homologs of the aforementioned proteins in the full UniprotKB database (version 2020_03), correlated the appearance of homologs in each species, and used the results to identify the spread of each type of nuclease at the order level (*Figure 6*). Orn was widespread in all eukaryotic groups, although in bacteria it was infrequently found outside of the Proteobacteria and Actinobacteria phyla. The structurally related RNase T is largely limited to Proteobacteria, with a few exceptions. Of the other major nucleases included in this analysis, NrnA was most frequently found in many bacteria. In contrast, NrnB occurs more sparsely distributed, potentially suggesting a more specialized function in individual organisms. NrnC is primarily found in most Alphaproteobacteria and Cyanobacteria, often overlapping with the occurrence of the homologous RNase D (with respect to the catalytic-domain sequence), although RNase D is present in many more bacterial genomes than NrnC. The three proteins that act as effective diribonucleases (Orn, NrnC, NrnA/NrnB) were largely – although not always – mutually exclusive; most bacterial taxa in this analysis had only one of the three, with the notable exception of Actinobacteria that frequently contained both Orn and NrnC (*Figure 6*).

We used the identified sequences of DnaQ family ribonucleases, each family curated individually, to create multiple sequence alignments and ultimately a combined phylogenetic tree of representative sequences of Orn, RNase T, NrnC, and RNase D family proteins (*Figure 7*). Similar to a previous

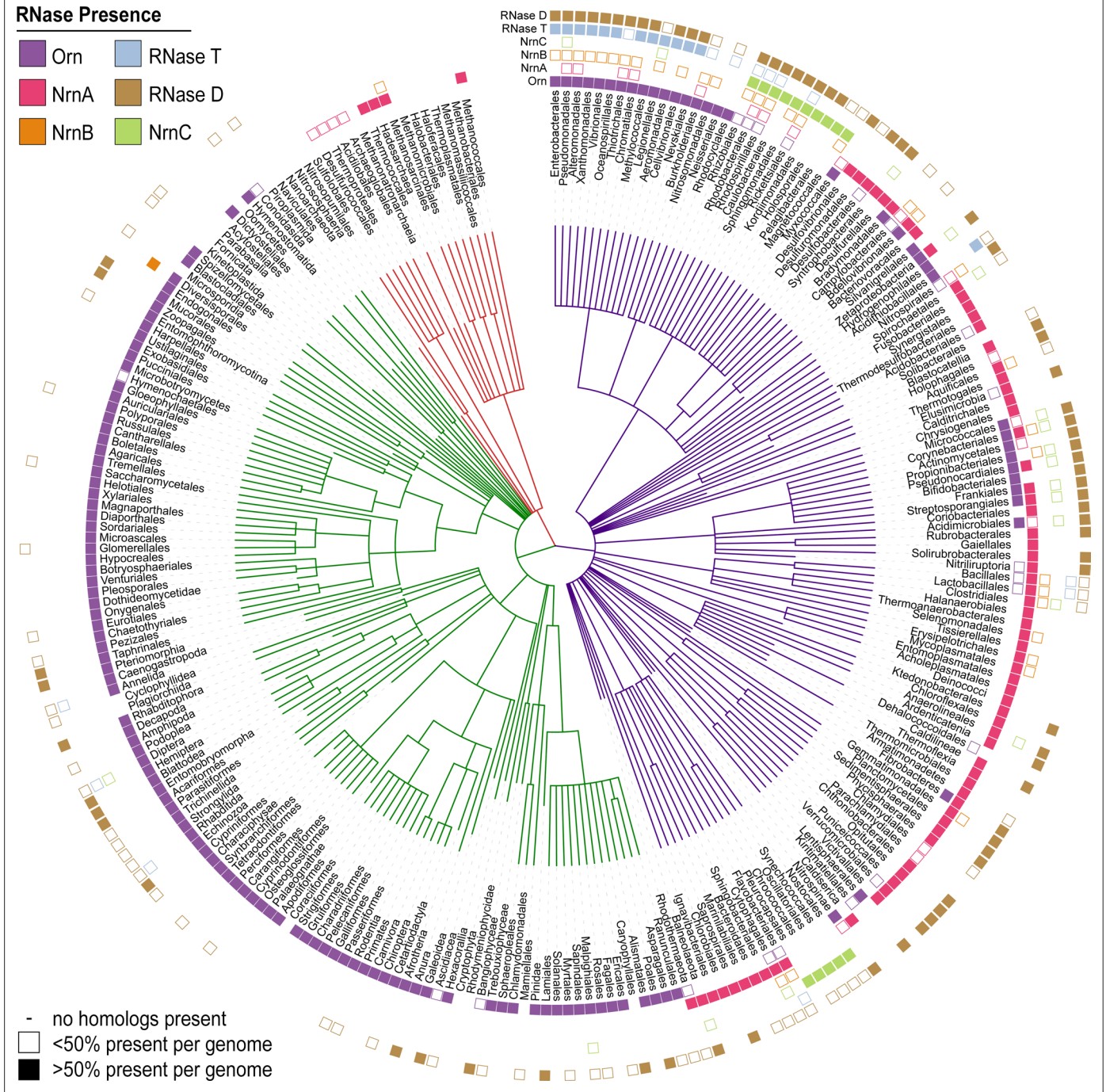

**Figure 6.** Presence of RNase homologs across sequenced organism classes. Shown is a 'Tree of Life' with all taxonomic groups at the class level with at least one substantially complete proteome available in the dataset. The tree is based on the structure of the NCBI Taxonomy database, with bacterial taxa shown with purple lines, eukaryotic taxa shown with green lines, and archaeal taxa shown with red lines. The presence of each RNase homolog as a proportion of the total proteins in that taxonomic group is shown as either a filled square (>50% presence of a homolog per genome) or an empty square (<50% presence of a homolog per genome). Lack of a square indicates no homologs for that family were present in genomes of that class.

analysis (*Zuo and Deutscher, 2001*), the DEDDh sequences (Orn and RNase T) segregated from the DEDDy sequences (RNase D and NrnC), highlighting the distinct evolutionary background of NrnC and Orn. Emanating from the ancestorial sequence, the first branch separated RNase T and Orn from RNase D and NrnC. This ancient split was soon followed by a split between RNase T and Orn. In the other lineage, NrnC and RNase D diverged from each other after a longer effective evolutionary time,

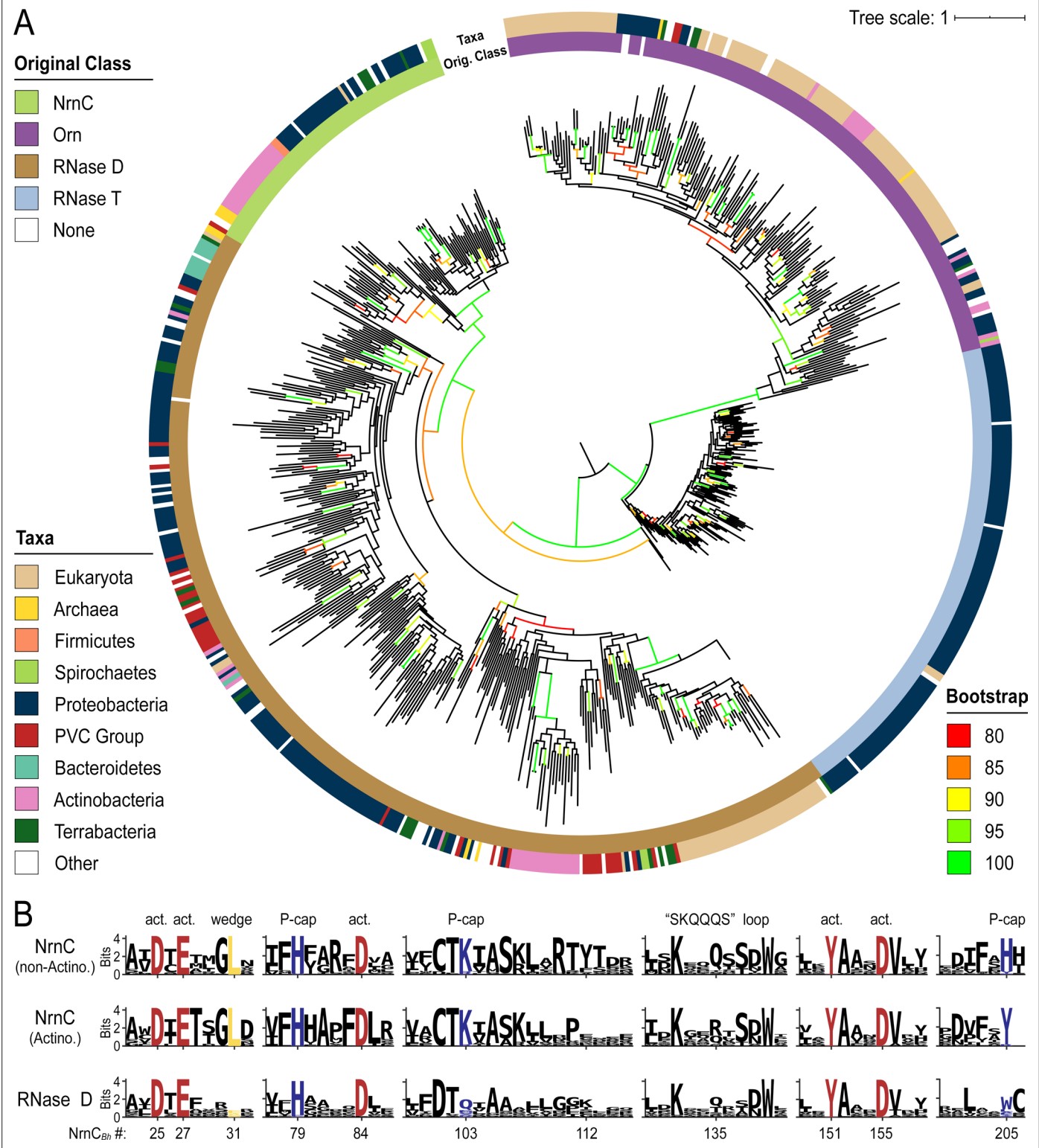

**Figure 7.** Phylogenetic tree of four DnaQ-fold RNase families. (**A**) Phylogenetic tree of 669 representatives of the RNase T, RNase D, oligoribonuclease (Orn), and nano-RNase C (NrnC) families of RNase proteins. The inner ring represents the original classification of each sequence by HMM analysis. The outer ring represents the high-level taxonomic classification of the organism the protein is found in. The color of the branch represents the UFBoot bootstrap value, where black branches are <80%, red is 80%, orange is 85%, yellow is 90%, light green is 95%, and bright green is 100% . Bootstrap values > 90% indicate high-confidence splits. (**B**) Sequence logos of RNase D and NrnC subgroups. Sequence logos showing the relative entropy (information content) at selected positions in RNase D as well as the Actinobacterial and non-Actinobacterial subsets of NrnC. Sequence numbering is relative to *Bartonella birtlessi* NrnC (G4VUY7). Active site residues are shown in red, phosphate cap residues in dark blue, and the L-wedge in yellow.

presumably following a duplication from the more closely related RNase D (*Figure 7A*). Due to this relatively recent split, and as it frequently co-occurs with RNase D, NrnC appears to have arisen from a more recent specialization event.

Catalytic, L-wedge, and phosphate cap residues are strictly conserved in many NrnC orthologs (*Figures 1D and 7B*). RNase D, the closest relative to NrnC, shares the catalytic DEDDy motif with NrnC, but lacks the L-wedge and phosphate cap (*Figure 7B*), suggesting that these features distinguish NrnC (and Orn) from other RNases and DnaQ-fold enzymes. The phylogenetic analysis also identified a unique group of NrnC-like enzymes in Actinobacteria, which shares most of the characteristic NrnC features, including conservation of the active site, most of the phosphate cap and wedge residues. However, subtle but specific changes in important residues (i.e., a Q-to-R change in the SKQQQS loop and the phosphate cap's $H^{205}$, which is replaced by a tyrosine residue) hint at the possibility of distinct function of the NrnC orthologs in this subgroup (*Figure 7*). Some Actinobacteria also encode an additional Orn and/or NrnA in the same genome, which could suggest either redundant functions or further specialization of these enzymes in these organisms (*Figure 6*).

## Conclusions

The establishment of NrnC as a dedicated dinuclease led us to compare and contrast the structural features of NrnC and Orn, the other enzyme with such a specific activity (*Kim et al., 2019*). Because

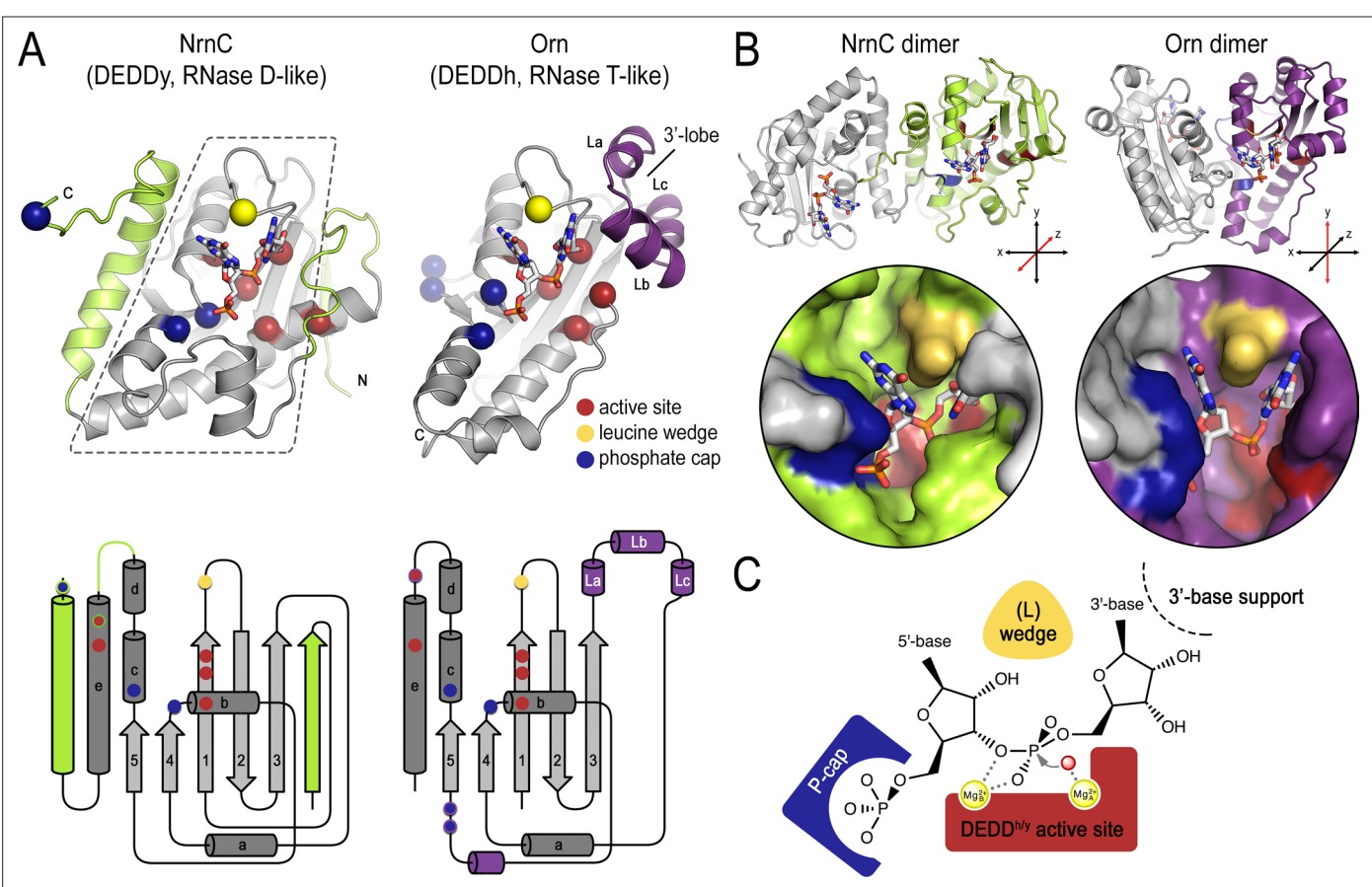

**Figure 8.** Structural comparison of nano-RNase C (NrnC) and oligoribonuclease (Orn). (**A**) Fold topology. pGG-bound NrnC and Orn monomers are shown in a similar orientation as cartoons (top) or schematic topology diagrams (bottom). Conserved catalytic core elements are colored in gray. NrnC and Orn-specific features are colored in green and purple, respectively. Other color codes mark the positions of the DEDDy/h motif (red spheres), L-wedge (yellow sphere), and phosphate cap residues (dark blue spheres). (**B**) Comparison of dimer units of NrnC and Orn (top) with close-ups of the composite active sites of the enzymes (bottom). An NrnC monomer is colored green and an Orn monomer is colored purple, with adjacent monomers in the biological assemblies colored in light gray. Specific residues are colored as in (**A**). Coordinate systems indicate the twofold symmetry axis of the enzyme dimers, with the colored monomers shown in a similar orientation. (**C**) Structurally and functionally conserved features common among NrnC- and Orn-type diribonucleases.

NrnC and Orn evolved independently from two different families of RNases, RNase D and RNase T, respectively, their shared substrate length specificity is particularly noteworthy. On the level of a single subunit, NrnC and Orn contain a conserved catalytic core comprising a β-sheet and adjacent α-helices, which harbors the residues for divalent cation coordination, the functionally important tyrosine or histidine residue ('DEDDy' or 'DEDDh'), and a wedge residue (L-wedge) that separates the substrate's bases (*Figure 8A*). In addition, the position of two residues contributing to the respective phosphate caps is conserved, but the identity of the residues varies between the two enzyme groups. For both enzyme families, the active site involves residues from two monomers requiring minimally a dimeric protein (*Figure 8B*). Thus, the two enzyme families are characterized by a functionally conserved active site optimized to accommodate the shortest RNA substrates, which distinguishes the dinucleases characterized to date from other RNases that act on longer substrates.

Despite these remarkable, function-defining commonalities between NrnC and Orn/Rexo2, their different evolutionary histories have led to distinct implementation of some of these important features. Particularly, NrnC and Orn differ in secondary structure motifs at the periphery of their conserved catalytic core. The enzyme family-specific parts include, for NrnC, additional phosphate cap residues, an N-terminal β-strand, and an additional C-terminal α-helix; and for Orn, short helices La, Lb, and Lc forming a lobe that coordinates the 3′ base of the dinucleotide substrate (*Figure 8A*). On the quaternary structure level, both NrnC and Orn form dimers (*Figure 8B*). However, the specific dimer arrangements vary between NrnC and Orn. The C-terminal α-helix of NrnC, which is absent in Orn, serves as the major dimerization interface in this family of enzymes, yielding a twofold-symmetric dimer. Orn and Orn-related enzymes (such as Rexo2) also form a twofold symmetric unit, but via a distinct rotation axis and the interface involves elements of the central core fold (*Figure 8B*). In contrast to Orn/Rexo2, whose biological unit is the dimeric form, four dimers of NrnC assemble further to the final octameric assembly. Lateral interfaces between dimers of NrnC within the octamer replace the 3′ lobe of Orn in coordinating the 3′ base of the substrate. Despite these major differences in the architectural composition and unique features of the two enzyme families, the elements that we identified as characteristic for dinucleases align nearly perfectly in space (*Figure 8B and C*).

Of the NrnC-specific motifs, the N-terminal β-strand and near-C-terminal α-helix are conserved in RNase D (*Figure 1—figure supplement 2B*). NrnC's C-terminal extension including H[205], placed in the active site of an adjacent subunit within a NrnC dimer, is involved in 5′ base stacking and completes the phosphate cap. Mutations in NrnC's unique phosphate cap retained residual activity, suggesting a role mainly in restricting the length of substrates accommodated at the active site (*Figure 2*). In contrast, corresponding mutations at the phosphate cap of Orn abolished catalytic activity, suggesting that these residues contribute directly to the catalytic mechanism in addition to imposing a substrate length restriction at the active site (*Kim et al., 2019*). One possibility is that the more ancient Orn evolved more stringent diribonucleotide preference compared to the more recently occurring NrnC-type enzymes. Taken together, the phosphate cap is a unifying feature of Orn and NrnC, though arisen independently in the two enzyme classes. In general, evolution of such specialized active sites that only degrade dinucleotides would also allow for rapid turnover of this specific nucleotide pool since competition from longer RNAs (or DNAs) would be suppressed. Whether similar motifs have evolved in other enzymes to restrict substrate length remains to be established.

Together, our bioinformatics analysis revealed the unique histories of NrnC and Orn, two nucleases that arose independently to fulfill the crucial role of dinucleotide degradation. Their activities are at the confluence of RNA metabolism and bacterial second messenger signaling, signifying their importance for cellular homeostasis and regulation. Indeed, functional NrnC or Orn, and consequently the clearance of the cellular dinucleotide pool, are necessary for proper growth, being essential in many organisms (*Ghosh and Deutscher, 1999*; *Kim et al., 2019*; *Liu et al., 2012*; *Sternon et al., 2018*), which could provide an avenue for targeted antimicrobial intervention. Specifically, as *Bartonella* and *Brucella* species are important pathogens (*Dehio, 2005*; *Greenfield et al., 2002*; *Pappas et al., 2006*), understanding the implications of NrnC function and failure could offer insight and effective strategies to battle the pernicious impacts of these organisms. Our structure-function studies present blueprints for such endeavors by revealing the specific active-site architectures and activity profiles of NrnC and Orn/Rexo2, their similarities and differences, as well as the general features that distinguish dinucleases from other 3′–5′ exonucleases.

# Materials and methods

**Key resources table**

| Reagent type (species) or resource | Designation | Source or reference | Identifiers | Additional information |
|---|---|---|---|---|
| Strain, strain background (*Pseudomonas aeruginosa*) | PA14 | *Rahme et al., 1995*; PMID:7604262 | | |
| Strain, strain background (*P. aeruginosa*) | PA14 Δ*orn* | This study | | |
| Strain, strain background (*Escherichia coli*) | Stellar cells | Takara/Clontech | | |
| Strain, strain background (*E. coli*) | BL21(DE3) | New England Biolabs | | |
| Recombinant DNA reagent | pEX-Gn-Δ*orn* (plasmid) | This study | | |
| Recombinant DNA reagent | pJN105 (plasmid) | *Newman and Fuqua, 1999*; PMID:10023058 | | |
| Recombinant DNA reagent | pJHA (plasmid) | *Kim et al., 2019*; PMID:31225796 | | |

*Continued on next page*

*Continued*

| Reagent type (species) or resource | Designation | Source or reference | Identifiers | Additional information |
|---|---|---|---|---|
| | ACAGATTGGTGGATC | | | |
| | CATGACCGAAATTCGCG | | | |
| | TGCATCAGGGCGATCTGC | | | |
| | CGAACCTGGATAACTAT | | | |
| | CGCATTGATG | | | |
| | CGGTGGCG | | | |
| | GTGGATACCG | | | |
| | AAACCCTGG | | | |
| | GCCTGCAGCC | | | |
| | GCATCGCGAT | | | |
| | CGCCTGTGCG | | | |
| | TGGTGCAGCTG | | | |
| | AGCAGCGGCGA | | | |
| | TGGCACCGC | | | |
| | GGATGTGATTCA | | | |
| | GATTGCGAA | | | |
| | AGGCCAGAAAA | | | |
| | GCGCGCCGAA | | | |
| | CCTGGTGCGCC | | | |
| | TGCTGAGCG | | | |
| | ATCGCGATATT | | | |
| | ACCAAAATTTTT | | | |
| | CATTTTGGCCGC | | | |
| | TTTGATCTG | | | |
| | GCGATTCTGGCG | | | |
| | CATACCTTTG | | | |
| | GCGTGATGCCG | | | |
| | GATGTGGTGT | | | |
| | TTTGCACCAAAAT | | | |
| | TGCGAGCAA | | | |
| | ACTGACCCGCAC | | | |
| | CTATACCGATCGC | | | |
| | CATGGCCTGAAAG | | | |
| | AAATTTGCGG | | | |
| | CGAACTGCTGAAC | | | |
| | GTGAACATTAG | | | |
| | CAAACAGCAGCAG | | | |
| | AGCAGCGATTG | | | |
| | GGCGGCGGAAAC | | | |
| | CCTGAGCCGCG | | | |
| | CGCAGATTGAATAT | | | |
| | GCGGCGAGCG | | | |
| | ATGTGCTGTATCTG | | | |
| | CATCGCCTGAA | | | |
| | AGATATTTTTGAAG | | | |
| | AACGCCTGAAA | | | |
| | CGCGAAGAACGCG | | | |
| | AAAGCGTGGCG | | | |
| | AAAGCGTGCTTTC | | | |
| | AGTTTCTGCCGA | | | |
| | TGCGCGCGAACC | | | |
| | TGGATCTGCTGG | | | |
| | GCTGGAGCGAAATTG | | | |
| | ATATTTTTGCG | | | |
| | CATAGCTAAGCGGC | | | Custom DNA fragment for *Bartonella henselae* |
| Sequence-based reagent | CGCACTCGAGCA (DNA fragment) | Geneart | BH02530 | NrnC, cloned in to His$_6$-SUMO-pET28 |

*Continued on next page*

*Continued*

| Reagent type (species) or resource | Designation | Source or reference | Identifiers | Additional information |
|---|---|---|---|---|
| Sequence-based reagent | ACAGATTGGTG GATCCATGACCA TTCGCTTTCATC GCAACGATCT GCCGAACCTGGA TAACTATCAGG TGGATGCGGTG GCGATTGATAC CGAAACCCTGG GCCTGAACCCGC ATCGCGATCGCC TGTGCGTGGTG CAGATTAGCCCG GGCGATGGCAC CGCGGATGTGA TTCAGATTGAAGC GGGCCAGAAAAAA GCGCCGAACC TGGTGAAACTGC TGAAAGATCGC AGCATTACCAAAA TTTTTCATTTTG GCCGCTTTGATC TGGCGGTGCTG GCGCATGCGTTT GGCACCATGCC GCAGCCGGTGTT TTGCACCAAAAT TGCGAGCAAACTG ACCCGCACCT ATACCGATCGCCAT GGCCTGAAAG AAATTTGCAGCGA ACTGCTGGATG TGAGCATTAGCAA ACAGCAGCAG AGCAGCGATTGGG CGGCGGAAG TGCTGAGCCAGG CGCAGCTGGAA TATGCGGCGAG CGATGTGCTGTAT CTGCATCGCCTG AAAGCGGTGCT GGAACAGCGCCT GGAACGCGAT GGCCGCACCAAA CAGGCGGAAGC GTGCTTTAAATTT CTGCCGACCCG CAGCGAACTGGA TCTGATGGGCTG GGCGGAAAGCGA TATTTTTGCGCAT AGCTAAGCGGCCGCACTC GAGCA (DNA fragment) | Geneart | BMEI1828 | Custom DNA fragment for *Brucella melitensis* NrnC, cloned in to His$_6$-SUMO-pET28 |
| Recombinant DNA reagent | His$_6$-SUMO-pET28-NrnC$_{Bh}$ (plasmid) | This study | | Cloned from custom DNA fragment |
| Recombinant DNA reagent | His$_6$-SUMO-pET28-NrnC$_{Bm}$ (plasmid) | This study | | Cloned from custom DNA fragment |
| Recombinant DNA reagent | pJHA-NrnC$_{Bh}$ (plasmid) | This study | | NrnC$_{Bh}$ cloned into pJHA for expression in *P. aeruginosa* |
| Recombinant DNA reagent | pJHA-NrnC$_{Bm}$ (plasmid) | This study | | NrnC$_{Bm}$ cloned into pJHA for expression in *P. aeruginosa* |
| Recombinant DNA reagent | pJHA-NrnC$_{Bh}$ D25A (plasmid) | This study | | Product of site-directed mutagenesis of pJHA-NrnC$_{Bh}$ |

*Continued*

| Reagent type (species) or resource | Designation | Source or reference | Identifiers | Additional information |
|---|---|---|---|---|
| Recombinant DNA reagent | pJHA-NrnC$_{Bh}$ E27A (plasmid) | This study | | Product of site-directed mutagenesis of pJHA-NrnC$_{Bh}$ |
| Recombinant DNA reagent | pJHA-NrnC$_{Bh}$ D84A (plasmid) | This study | | Product of site-directed mutagenesis of pJHA-NrnC$_{Bh}$ |
| Recombinant DNA reagent | pJHA-NrnC$_{Bh}$ D155A (plasmid) | This study | | Product of site-directed mutagenesis of pJHA-NrnC$_{Bh}$ |
| Recombinant DNA reagent | pJHA-NrnC$_{Bh}$ Y151A (plasmid) | This study | | Product of site-directed mutagenesis of pJHA-NrnC$_{Bh}$ |
| Recombinant DNA reagent | pJHA-NrnC$_{Bh}$ L31A (plasmid) | This study | | Product of site-directed mutagenesis of pJHA-NrnC$_{Bh}$ |
| Recombinant DNA reagent | pJHA-NrnC$_{Bh}$ H79A (plasmid) | This study | | Product of site-directed mutagenesis of pJHA-NrnC$_{Bh}$ |
| Recombinant DNA reagent | pJHA-NrnC$_{Bh}$ K103A (plasmid) | This study | | Product of site-directed mutagenesis of pJHA-NrnC$_{Bh}$ |
| Recombinant DNA reagent | pJHA-NrnC$_{Bh}$ H205A (plasmid) | This study | | Product of site-directed mutagenesis of pJHA-NrnC$_{Bh}$ |
| Recombinant DNA reagent | pJHA-NrnC$_{Bh}$ K132A (plasmid) | This study | | Product of site-directed mutagenesis of pJHA-NrnC$_{Bh}$ |
| Sequence-based reagent | TTTGGGCTAGCCA TATGACCGAAATT CGTGTTCATCAGGG | Life Technologies | NrnC$_{Bh}$_infusionprimer_F | Primer for cloning NrnC$_{Bh}$ into pJHA |
| Sequence-based reagent | GCTCAAGCTTGAAT TCGCTGTGTGCAA AGATATCAATTTCG | Life Technologies | NrnC$_{Bh}$_infusionprimer_R | Primer for cloning NrnC$_{Bh}$ into pJHA |
| Sequence-based reagent | TTTGGGCTAGCCA TATGACCATTCGTT TTCATCGTAATGATC | Life Technologies | NrnC$_{Bm}$_infusionprimer_F | Primer for cloning NrnC$_{Bm}$ into pJHA |
| Sequence-based reagent | GCTCAAGCTTG AATTCGCTATGTG CAAAAATATCGCTTTC | Life Technologies | NrnC$_{Bm}$_infusionprimer_R | Primer for cloning NrnC$_{Bm}$ into pJHA |
| Sequence-based reagent | Acccagtgtttcggtagc aacggcaactgcatc | Life Technologies | NrnC$_{Bh}$ D25A_a | Primer for site directed mutagenesis |
| Sequence-based reagent | Gatgcagttgccgttgct accgaaacactgggt | Life Technologies | NrnC$_{Bh}$ D25A_b | Primer for site directed mutagenesis |
| Sequence-based reagent | Gttgccgttgataccgca acactgggtctgcag | Life Technologies | NrnC$_{Bh}$ E27A_a | Primer for site directed mutagenesis |
| Sequence-based reagent | Ctgcagacccagtgttgc ggtatcaacggcaac | Life Technologies | NrnC$_{Bh}$ E27A_b | Primer for site directed mutagenesis |

*Continued*

| Reagent type (species) or resource | Designation | Source or reference | Identifiers | Additional information |
|---|---|---|---|---|
| Sequence-based reagent | Cgatgcggctgcgcac ccagtgtttcggtatcaacg | Life Technologies | NrnC$_{Bh}$ L31A_a | Primer for site directed mutagenesis |
| Sequence-based reagent | Cgttgataccgaaaca ctgggtgcgcagc cgcatcg | Life Technologies | NrnC$_{Bh}$ L31A_b | Primer for site directed mutagenesis |
| Sequence-based reagent | Agatcgaaacgacca aaggcaaagattttg gtaatatcacgatcgc | Life Technologies | NrnC$_{Bh}$ H79A_a | Primer for site directed mutagenesis |
| Sequence-based reagent | Gcgatcgtgatattacc aaaatctttgcctttg gtcgtttcgatct | Life Technologies | NrnC$_{Bh}$ H79A_b | Primer for site directed mutagenesis |
| Sequence-based reagent | Gtgccagaattgccag agcgaaacgac caaagtga | Life Technologies | NrnC$_{Bh}$ D84A_a | Primer for site directed mutagenesis |
| Sequence-based reagent | Tcactttggtcgtttcgct ctggcaattctggcac | Life Technologies | NrnC$_{Bh}$ D84A_b | Primer for site directed mutagenesis |
| Sequence-based reagent | Cgggtcagtttgcttgc aattgcggtacaa aaaacaacatccgg | Life Technologies | NrnC$_{Bh}$ K103A_a | Primer for site directed mutagenesis |
| Sequence-based reagent | Ccggatgttgttttttgtacc gcaattgcaagca aactgacccg | Life Technologies | NrnC$_{Bh}$ K103A_b | Primer for site directed mutagenesis |
| Sequence-based reagent | Acatcacttgctgcagct tcaatctgtgcac ggctcagg | Life Technologies | NrnC$_{Bh}$ Y151A_a | Primer for site directed mutagenesis |
| Sequence-based reagent | Cctgagccgtgcacag attgaagctgca gcaagtgatgt | Life Technologies | NrnC$_{Bh}$ Y151A_b | Primer for site directed mutagenesis |
| Sequence-based reagent | Cggtgcagatacaga acagcacttgctgcatattcaa | Life Technologies | NrnC$_{Bh}$ D155A_a | Primer for site directed mutagenesis |
| Sequence-based reagent | Ttgaatatgcagcaag tgctgttctgtatctgcaccg | Life Technologies | NrnC$_{Bh}$ D155A_b | Primer for site directed mutagenesis |
| Sequence-based reagent | Gctcaagcttgaattc gctggctgcaa agatatcaatttcgc | Life Technologies | NrnC$_{Bh}$ H205A_a_pJHA | Primer for site directed mutagenesis |
| Sequence-based reagent | Gcgaaattgatatctttgc agccagcgaattcaagcttgagc | Life Technologies | NrnC$_{Bh}$ H205A_b_pJHA | Primer for site directed mutagenesis |
| Sequence-based reagent | Gtgcggccgcttagctgg ctgcaaagatatcaatttcgct | Life Technologies | NrnC$_{Bh}$ H205A_a_SUMO | Primer for site directed mutagenesis |
| Sequence-based reagent | Agcgaaattgatatcttt gcagccagctaa gcggccgcac | Life Technologies | NrnC$_{Bh}$ H205A_b_SUMO | Primer for site directed mutagenesis |
| Sequence-based reagent | 5'-GG-3' (RNA primer) | Sigma | | |
| Sequence-based reagent | 5'-AGG-3' (RNA primer) | Sigma | | |
| Sequence-based reagent | 5'-AAGG-3' (RNA primer) | Sigma | | |
| Sequence-based reagent | 5'-AAAGG-3' (RNA primer) | Sigma | | |
| Sequence-based reagent | 5'-AAAAGG-3' (RNA primer) | Sigma | | |

*Continued*

| Reagent type (species) or resource | Designation | Source or reference | Identifiers | Additional information |
|---|---|---|---|---|
| Sequence-based reagent | 5′-AAAAAGG-3′ (RNA primer) | Sigma | | |
| Sequence-based reagent | 5′-pGG-3′ (RNA primer) | Biolog' catalog number P023-01 | | |
| Sequence-based reagent | 5′-pAA-3′ (RNA primer) | Biolog' catalog number P033-01 | | |
| Sequence-based reagent | 5′-pGC-3′ (RNA primer) | GE Healthcare Dharmacon | | |
| Chemical compound, drug | 5′-pAp-3′ (RNA primer) | Sigma | Cat# A5763 | |
| Sequence-based reagent | Various RNA and DNA oligonucleotides | IDT | | |
| Antibody | Anti-HA (rabbit polyclonal) | Takara | Cat# 631207 | (1:100) |
| Antibody | Anti-HA–agarose (mouse monoclonal, clone HA-7) | Sigma | Cat# A2095; RRID:AB_257974 | (10 µl) |
| Antibody | Anti-rabbit (donkey polyclonal, HRP-conjugated) | Cytiva | Cat# NA934 | (1:5000) |
| Software, algorithm | Prism | GraphPad | RRID:SCR_002798 | |
| Software, algorithm | XDS Program Package | *Kabsch, 2010*; PMID:20124693 | RRID:SCR_015652 | Distributed through SBGrid |
| Software, algorithm | Pointless | *Evans, 2006*; PMID:16369096 | RRID:SCR_014218 | Distributed through SBGrid |
| Software, algorithm | Scala | *Evans, 2006*; PMID:16369096 | | Distributed through SBGrid |
| Software, algorithm | Phenix | *Adams et al., 2010*; PMID:20124702 | RRID:SCR_014224 | Distributed through SBGrid |
| Software, algorithm | Coot | *Emsley et al., 2010*; PMID:20383002 | RRID:SCR_014222 | Distributed through SBGrid |
| Software, algorithm | MrBump | *Keegan et al., 2018*; PMID:29533225 | | Distributed through SBGrid |
| Software, algorithm | PyMOL | Schrödinger | RRID:SCR_000305 | Distributed through SBGrid |
| Software, algorithm | UCSF ChimeraX | *Pettersen et al., 2021*; PMID:32881101 | RRID:SCR_015872 | Distributed through SBGrid |
| Software, algorithm | cryoSPARC | *Punjani et al., 2017*; PMID:28165473 | RRID:SCR_016501 | |
| Software, algorithm | RELION | *Zivanov et al., 2018*; PMID:30412051 | RRID:SCR_016274 | |
| Software, algorithm | GCTF | *Zhang, 2016*; PMID:26592709 | RRID:SCR_016500 | |
| Software, algorithm | iTOL | *Letunic and Bork, 2019*; PMID:30931475 | RRID:SCR_018174 | |
| Software, algorithm | TCoffee | *Notredame et al., 2000*; PMID:10964570 | RRID:SCR_019024 | |
| Software, algorithm | Hmmer | *Eddy, 2011*; PMID:22039361 | RRID:SCR_005305 | |

Continued

| Reagent type (species) or resource | Designation | Source or reference | Identifiers | Additional information |
|---|---|---|---|---|
| Software, algorithm | MAFFT | *Katoh et al., 2005*; PMID:15661851 | RRID:SCR_011811 | |
| Software, algorithm | SnakeMake | *Köster and Rahmann, 2018*; PMID:29788404 | RRID:SCR_003475 | |

## Expression constructs and mutagenesis

For protein expression in *Escherichia coli*, codon-optimized NrnC genes from *B. henselae* (BH02530) and *B. melitensis* (BMEI1828) were synthesized by Geneart (Life Technologies). Genes were cloned into a modified pET28a vector (Novagen) between BamHI and NotI sites using InFusion cloning (Takara Bio). The resulting fusion proteins expressed from these plasmids contained an N-terminal His$_6$-tagged small ubiquitin-like modifier (SUMO) cleavable by recombinant Ulp-1 protease.

For the arabinose-inducible expression and detection of NrnC in *P. aeruginosa*, we used a modified pJN105 vector (*Newman and Fuqua, 1999*). The vector pJGHA was constructed by inserting a coding sequence for monomeric superfolder GFP (msfGFP)-HA epitope fusion between the NheI and XbaI sites in pJN105. The plasmid allows insertion of genes of interest between novel NdeI and EcoRI sites, and their expression results in proteins with C-terminal msfGFP-HA. The vector pJHA was constructed by digesting pJGHA with HindIII (New England Biolabs) to remove the msfGFP coding sequence. Following re-ligation of the gel-purified restriction digest, the coding sequence for the HA epitope remained, allowing proteins of interest to be expressed with a C-terminal HA epitope.

For expression in *P. aeruginosa*, the codon-optimized NrnC$_{Bh}$ sequences were amplified by PCR and inserted between NdeI and EcoRI sites of the pJHA vector using InFusion cloning.

A QuikChange II site-directed mutagenesis kit (Agilent) was used for the introduction of point mutations in *nrnc$_{Bh}$* following the manufacturer's instructions. All mutations were verified by DNA sequencing.

## Protein expression and purification

*E. coli* BL21 T7 Express cells (New England Biolabs) were transformed with pET28a plasmids encoding His$_6$-SUMO-NrnC$_{Bh}$ or -NrnC$_{Bm}$ and grown in Terrific Broth (TB) supplemented with 50 μg/ml kanamycin at 37 °C to an OD$_{600}$ of ~1.0. Induction was carried out at 18 °C with 0.5 mM IPTG for 16 hr. Cells were harvested by centrifugation, resuspended in minimal volume of Ni-NTA binding buffer (25 mM Tris-Cl pH 8.5, 500 mM NaCl, 20 mM imidazole), frozen in liquid nitrogen, and stored at –80 °C.

Cell pellets were thawed followed by lysis through sonication and centrifugation. Supernatants were incubated on ice with Ni-NTA resin (Qiagen) equilibrated with Ni-NTA binding buffer for 1 hr with gentle agitation. The NrnC-bound resin was washed three times with 10 column volumes of Ni-NTA binding buffer by gravity flow. Bound NrnC was eluted in six-column volumes of Ni-NTA elution buffer (25 mM Tris-Cl pH 8.5, 500 mM NaCl, 400 mM imidazole). Eluates were buffer exchanged into gel filtration buffer (25 mM Tris-Cl, 150 mM NaCl, pH 7.5) via a HiPrep 26/10 desalting column (GE Healthcare) and incubated overnight with Ulp-1-His$_6$ to cleave the His$_6$-tagged SUMO moiety from NrnC. Following Ulp-1 cleavage, untagged NrnC protein was recovered in the flow through of a HisTrap Ni-NTA column (GE Healthcare), separated from His$_6$-SUMO, and ULP1-His$_6$. EDTA at a final concentration of 10 mM was added to NrnC before concentration via an Amicon Ultra 10K concentrator (Merck Millipore). Concentrated NrnC was injected onto a HiLoad 16/60 Superdex 200 gel filtration column (GE Healthcare) equilibrated in gel filtration buffer. Fractions containing NrnC were concentrated, frozen in liquid nitrogen, and stored at –80 °C.

## Crystallization, data collection, and structure refinement

NrnC-RNA complexes (pGG and pAA from Biolog Life Science Institute, other nucleotides from Dharmacon) were formed prior to crystallization by mixing 1:2 molar ratio of protein:nucleotide in gel filtration buffer, followed by 30 min incubation at the crystallization temperature. Protein concentrations used in crystallization ranged from 2.5 to 10 mg/ml (NrnC$_{Bh}$) and 1.0 to 8.0 mg/ml (NrnC$_{Bm}$). Crystals were grown via hanging-drop vapor diffusion by mixing equal volumes (0.8 μl) of sample with reservoir solution. NrnC$_{Bh}$ crystals grew at 20 °C over a reservoir solution containing 0.1 M succinic acid (pH 6.5), 15–20% PEG 3350, and 20% xylitol. NrnC$_{Bm}$ crystals grew at 4°C and 20°C, in reservoir

solutions containing 0.1 M Tris-Cl (pH 7), 2.0–2.4 M ammonium sulfate or 1.4 M sodium-potassium phosphate, and 20% xylitol. Prior to freezing crystals in liquid nitrogen, they were soaked in reservoir solution with up to 25% xylitol. Data were collected by synchrotron radiation on frozen crystals at Cornell High Energy Synchrotron Source (CHESS) and NE-CAT 24ID-C and 24ID-E beamlines at the Advanced Photon Source (APS) at Argonne National Laboratory. Diffraction data sets were processed using XDS, Pointless, and Scala (*Evans, 2006*; *Kabsch, 2010*). The initial structures were solved by molecular replacement using the software package Phenix (*Adams et al., 2010*) and MrBUMP in the ccp4 suite (*Keegan et al., 2018*; *Winn et al., 2011*) with the coordinates of *E. coli* RNase D (PDB:1yt3, *Zuo et al., 2005*) as the search model. Manual model building and refinement were carried out with Coot (*Emsley et al., 2010*) and Phenix. Illustrations were prepared in Pymol (version 2.0 Schrödinger, LLC). All software packages were accessed through SBGrid (*Morin et al., 2013*). All data collection and refinement statistics are summarized in *Supplementary file 1*.

## Structure determination by cryo-EM

Purified NrnC$_{Bh}$ was diluted in buffer (25 mM Tris-Cl pH 7.5, 150 mM NaCl) to 4, 5, or 6 mg/ml (for incubation with pGG, pAGG, and pAAAGG, respectively). RNA substrate was added at threefold molar excess. After 15 min at room temperature, NP40 was added at 0.01% v/v and samples were placed on ice for an additional 15 min. Alternatively, NrnC was diluted to 7.5 mg/ml and incubated with threefold excess pAAAGG; after 15 min, 1/5 volume of buffer with CaCl$_2$ and NP40 (25 mM Tris-Cl pH 7.5, 150 mM NaCl, 5 mM CaCl$_2$, 0.05% NP40) was added and incubations proceeded for another 15 min on ice. All cryo-EM samples were prepared with Quantifoil R1.2/1.3 300-mesh grids after glow discharging in a PELCO easiGlow (60 s glow, 10 mA current, 80% Ar/20% O$_2$) using a FEI Mark IV Vitrobot (4 °C, 100% humidity, 2.5 s blot) to plunge grids into liquid nitrogen-cooled ethane immediately after blotting. Data were collected using the Cornell CCMR facility Thermo Fisher Scientific Talos Arctica with a Gatan K3 detector and BioQuantum energy filter operated at 200 kV at a nominal magnification of 63 kX (1.24 Å/pixel), 20 eV slit width, and 0.5× binning (super-resolution). Movies were collected with a total dose of 50 e/Å$^2$, fractionated into either 50 or 75 frames.

Data processing was performed using RELION 3.1 (*Zivanov et al., 2018*) and cryoSPARC (*Punjani et al., 2017*). Super-resolution movie exposures were aligned, dose-weighted, and Fourier-cropped to the physical pixel size in RELION using MotionCor2 (*Zheng et al., 2017*), and defocus values were estimated using GCTF (*Zhang, 2016*). Micrographs were then imported into cryoSPARC for manual curation, particle picking, and classification. Particles were picked using the cryoSPARC 'blob' and template picking and initially extracted with Fourier-cropping to a nominal pixel size of 2.89 Å. This particle stack was cleaned with 2D and 3D classifications in cryoSPARC, then re-extracted in RELION (1.24 Å/pixel) for 3D refinement, CTF refinement, and Bayesian polishing. Further 2D and 3D classification of CTF-refined particles in RELION was performed for the pAGG dataset. For each dataset, no symmetry was imposed during processing until a final refinement was performed imposing D4 symmetry. The crystal structure of NrnC$_{Bh}$ was docked into the reconstructed cryo-EM density maps using the program package Phenix (*Liebschner et al., 2019*) and the models were refined in Coot (*Emsley et al., 2010*), ChimeraX (*Pettersen et al., 2021*), and the real-space refinement module for cryo-EM in Phenix (*Afonine et al., 2018*). Illustrations were prepared with ChimeraX and show the density of the sharpened maps. All data collection and refinement statistics are summarized in *Supplementary file 2*.

## Complementation analysis in *P. aeruginosa*

Deletion of *orn* in *P. aeruginosa* UCBPP-PA14 was performed using two-step allelic exchange as described by Hmelo and colleagues (*Hmelo et al., 2015*). Briefly, deletion alleles were created by overlap extension PCR, and delivered on a pEX18 suicide plasmid to the *P. aeruginosa* host strain by conjugation with *E. coli* donor strain S17.1, leading to the removal of the gene from the genome. To test for complementation, genes were introduced into the *P. aeruginosa* *orn* deletion strains by using electroporation (*Choi et al., 2006*). Briefly, *P. aeruginosa* cells were grown overnight, centrifuged, then washed with and resuspended in 300 mM sucrose. Expression plasmids based on the pJHA vector were mixed with 100 µl of resuspended cells and electroporated using a Micropulser (Bio-Rad) followed by recovery in 1 ml of lysogeny broth (LB), shaking at 250 rpm for 1 hr at 37 °C. Recovered

cells were plated on LB plates containing 60 μg/ml gentamicin. Individual colonies were used for subsequent experiments.

## Drip assay

The indicated *P. aeruginosa* strains harboring expression plasmids were grown overnight with shaking at 37 °C in LB supplemented with 60 μg/ml gentamicin. The cells were adjusted to CFU = 10,000 in fresh LB medium and applied to LB plates supplemented with 60 μg/ml gentamicin and 0.2% arabinose in 20 μl drops. The plates were inverted allowing the culture to drip down the length of the plate, followed by incubation overnight at 37 °C. The plates were imaged using a Chemidoc MP imager (Bio-Rad) with a 0.2 s exposure time.

The colony-measurer Python program (https://github.com/gwmarrah/colony-measurer; *Marrah, 2021*) was employed to quantify the size of bacterial colonies by pixel measurement. Images were prepared for size quantification by cropping each lane of a drip plate as an individual 8-bit image. ImageJ (*Schneider et al., 2012*) and colSizeMeasurer.py were used to determine the background pixel intensity and minimum/maximum colony sizes to be measured, respectively. These values were used to refine the parameters in colSizeAnalyzer.py for accurate measurement.

## Immunoprecipitation and western blot

*P. aeruginosa* strains containing plasmid-borne NrnC$_{Bh}$-HA were grown overnight, followed by dilution to an $OD_{600}$ = 0.1 in fresh LB supplemented with 60 μg/ml gentamicin. Cultures were allowed to grow to an $OD_{600}$ = 0.8. Arabinose was added to a final concentration of 0.2% to induce protein expression for 2 hr at 37 °C. Following induction, cultures were normalized by OD, pelleted, and flash frozen in liquid nitrogen. Cells were resuspended in lysis buffer (150 mM NaCl, 25 mM Tris-Cl pH 7.5) followed by sonication. Anti-HA resin (Sigma) was prewashed with lysis buffer. Resin was added to the cleared lysate and incubated with rotation for 1 hr at 4 °C. Following binding, the HA resin was washed with lysis buffer, boiled in SDS loading buffer, and resolved by SDS-PAGE. Western blot transfer to a PVDF membrane proceeded for 90 min at constant 0.25 A, followed by overnight blocking with superblock (ThermoFisher) at 4 °C. Rabbit anti-HA primary antibody (Takara Bio) was diluted to 1:100 in TBS-T and incubated with the membrane for 1 hr at 20 °C. Following washes with TBS-T, an HRP-conjugated, anti-rabbit antibody (GE Life Sciences) was diluted to 1:5000 in TBS-T and incubated with the membrane for 30 min at 20 °C. The membrane was washed with TBS-T before treating with SuperSignal West Femto (ThermoFisher) ECL reagent, followed by imaging with a Bio-Rad Chemidoc system.

## Size-exclusion chromatography-coupled multiangle light scattering (SEC-MALS)

Purified, wild-type or mutant variant NrnC$_{Bh}$ at 2 mg/ml (85 μM) was injected onto a Superdex 200 Increase 10/300 gel filtration column (GE Healthcare) equilibrated with gel filtration buffer (25 mM Tris-Cl, pH 7.5, 150 mM NaCl). Size-exclusion chromatography was coupled to an in-line, static 18-angle light scattering detector (DAWN HELEOS-II, Wyatt Technology) and a refractive index detector (Optilab T-rEX, Wyatt Technology). Data were collected every second. Data analysis was performed with Astra 6.1 (Wyatt Technology) yielding the molar mass and mass distribution (polydispersity) of the sample. Monomeric BSA (Sigma; 5 mg/ml) was used as a control sample and to normalize the light scattering detectors.

## Measuring dissociation constant ($K_d$) and binding specificity by DraCALA

To measure $K_d$, the purified protein was serially diluted in binding buffer (10 mM Tris-HCl, pH 8, 100 mM NaCl, and 5 mM CaCl$_2$). Each dilution was mixed with $^{32}$P-labeled substrate and spotted onto nitrocellulose. The dried nitrocellulose was exposed to a phosphorimager screens, scanned, and analyzed as previously described (*Roelofs et al., 2011*). The fraction bound was plotted against protein concentration using the program Prism. For competition experiments to determine binding specificity, 100 μM of unlabeled nucleotides were mixed with $^{32}$P-labeled pGG; the mixtures were added 200 nM of purified NrnC.

## Biochemical assay of RNase activity

The reactions were performed by adding the indicated concentration of enzyme to the indicated concentration of substrate containing a tracer, consisting of 5′-end $^{32}$P-labeled substrate with the same length and sequence, in reaction buffer (10 mM Tris-HCl, pH 8, 100 mM NaCl, and 5 mM MgCl$_2$). Reactions are stopped at indicated time points by the addition of 0.2 M EDTA. The samples were mixed with loading buffer (4 M urea, 20% sucrose, 0.1% SDS, 0.05% bromophenol blue, 0.05% xylene cyanole FF, and 1× TBE), and separated by electrophoresis on 20% polyacrylamide gels.

## DNase activity measurement

DNase activity was assessed using an unspecific 1.5 kb PCR fragment, either with blunt ends or after restriction digestion with either KpnI, NdeI, or NotI (New England Biolabs). CIP (New England Biolabs) was used to dephosphorylate overhangs, and PNK (New England Biolabs) was used to phosphory-late blunt PCR products. NaCl, MgCl$_2$, and MnCl$_2$ were added to the concentration indicated in the figures, with concentrations of NrnC and DNA at 1 µM and 20 nM, respectively, except where indi-cated otherwise in the figure. Reactions were incubated for 30 min or the indicated time at 37 °C. Reactions were stopped by the addition of a stop buffer containing proteinase K (Qiagen) and EDTA (JT Baker) to a final concentration of 0.1 mAU proteinase K and 10 mM EDTA, followed by incubation at 50 °C for 30 min. Samples were resolved by electrophoresis in a 1% agarose gel containing GelRed stain (Biotium) and imaged by UV visualization in a GelDocXR system (Bio-Rad).

## Degradation of oligonucleotides by NrnC

5′-Hydroxyl DNA and RNA oligonucleotides were purchased from Integrated DNA Technologies (IDT). Indicated DNA and RNA oligonucleotides were 5′end-radiolabeled with $^{32}$P γ-ATP using T4 Polynu-cleotide Kinase. The radiolabeled DNA or RNA oligonucleotides were annealed with the complemen-tary DNA or RNA, respectively, by heating at 95 °C for 10 min in a heat block followed by removal of the heat block from the heat source and slow cooling to room temperature. NrnC (5 nM) was added to $^{32}$P-labeled substrates (3.3 nM) in reaction buffer. Reactions were stopped at indicated times. Short RNA or DNA oligos were analyzed by polyethyleneimine-cellulose TLC plates. The TLC plates were developed by saturated NH$_4$SO$_4$ and 1.5 M KH$_2$PO$_4$. For longer annealed DNA or RNA oligonucleotides, the reactions were separated by 20% urea PAGE, imaged using a Fujifilm FLA-7000 phosphorimager.

## Identification of RNase homologs

For each group of RNases, a list of seed protein sequences was manually prepared with Uniprot entry names: Orn (ORN_PSEAE, ORN_ECOLI, ORN_HUMAN, ORN_STRGR, ORN_CORDI, ORN_BURMA, ORN_YEAST, ORN_VIBCH), NrnA (NRNA_BACSU, NRNA_MYCPN, NRNA_THET8, NRNA_MYCTU, A0A3R9HUU0_STRSA), NrnB (NRNB_BACSU, A0A5C5X7X8_9BACT), NrnC (A9CG28_AGRFC, G4VUY7_9RHIZ, A1UU18_BARBK), RNase T (RNT_ECOLI, RNT_VIBCH, RNT_BUCAP, RNT_HAEIN, RNT_XYLFA), and RNase D (RND_ECOLI, Q9ZD81_RICPR, RND_HAEIN, I6XF17_MYCTU).

Initially a seed alignment was prepared for each RNase family using T-COFFEE (*Notredame et al., 2000*) in the Expresso structural alignment mode (*Armougom et al., 2006*). A search on the 2020_03 release of UniprotKB was performed with an HMM prepared from these alignments with hmmsearch from HMMER 3.3 (*Eddy, 2011*). Results were filtered by two criteria, a hit score above 125 and a template/query length ratio between 0.8 and 1.2. Search hits were used to construct a new multiple sequence alignment with MAFFT in E-INS-i mode (*Katoh et al., 2005*). The hmmsearch was repeated with an HMM constructed from this expanded alignment and results were filtered with the same hit score cutoff but a more generous template/query length ratio between 0.6 and 1.5. The resulting hits were considered to be the final list for each RNase family, with any sequences found in multiple cate-gories assigned to the category for which it had a higher hit score.

To determine whether a particular taxa contained homologs for each RNase group, the total number of proteins for each taxon in the NCBI taxon database present in our dataset was counted (*Federhen, 2012*). The average number of proteins per genome for each taxon was determined for all genomes annotated as a descendent taxon available in the NCBI genome database. Finally, the number of homologs found in each taxonomic category was calculated as a fraction of the total proteins in that taxon, multiplied by the average genome size to get an average presence-per-genome. For

visualization purposes, values above 0.5 are considered present and non-zero values below 0.5 are potentially or partially present. The minimal species tree was extracted from the NCBI taxonomy database using ETE3 (*Huerta-Cepas et al., 2016*), followed by visualization of the resulting tree and annotation with iTOL (*Letunic and Bork, 2019*).

All commands and code from this analysis were constructed as a reproducible SnakeMake pipeline (*Köster and Rahmann, 2018*) available at https://github.com/jgoodson/rnases, (copy archived at swh:1:rev:66b266487bfafb3dfc8917dbfd710c9ef7dc0220;*Goodson, 2021 Goodson, 2021*) (commit 66b2664 used in current versions of the figures).

## Phylogenetics of DnaQ-family RNases

To construct a phylogenetic tree of the DnaQ-fold RNases, the final sequences identified from the previous analysis for each family were clustered by sequence identity with MMSeqs2 (*Steinegger and Söding, 2017*). Targeting a final sequence count of 600, the sequence identity threshold was determined for each family necessary to approximately maintain the original proportion of each family in the final representatives (30% for Orn, 45% for NrnC, 50% for RNase T, and 30% for RNase D). From these cluster representatives, a multiple sequence alignment was constructed using MAFFT in E-INS-i mode using DASH to obtain additional structural homologs (*Rozewicki et al., 2019*). The alignment was trimmed by removing the additional DASH sequences and columns with more than 90% gaps. The most appropriate evolutionary model was determined with IQ-TREE ModelFinder (LG + R8) and a phylogenetic tree was constructed using IQ-TREE 2.1.1 with 10,000 UFBoot replicates, and some modified parameters for expanded tree search (additional UFBoot NNI search, initial SPR search radius 20, 500 initial trees, initial search on 100 best trees, maintenance of the 50 best trees, and 500 iterations without improvement as stopping criteria) (*Hoang et al., 2018*; *Kalyaanamoorthy et al., 2017*; *Minh et al., 2020*). The tree was rooted with midpoint rooting on the long internal branch between RNaseT/Orn and RNaseD/NrnC. Sequence logos were created from monophyletic subgroup alignments using Logomaker (*Tareen and Kinney, 2020*).

## Data deposition

The atomic coordinates and structure factors have been deposited in the Protein Data Bank, https://www.rcsb.org/ (PDB ID codes 7MPL, 7MPM, 7MPN, 7MPO, 7MPP, 7MPQ, 7MPR, 7MPS, 7MPT, 7MPU, 7MQB/EMD-23941, 7MQD/EMD-23943, 7MQF/EMD-23945, 7MQH/EMD-23947, 7MQC/EMD-23942, 7MQE/EMD-23944, 7MQG/EMD-23946, 7MQI/EMD-23948).

## Acknowledgements

This work is based upon research conducted at the Northeastern Collaborative Access Team beamlines, which are funded by the National Institute of General Medical Sciences from the National Institutes of Health (P30 GM124165). The Eiger 16 M detector on 24-ID-E is funded by a NIH-ORIP HEI grant (S10OD021527). This research used resources of the Advanced Photon Source, a US Department of Energy (DOE) Office of Science User Facility operated for the DOE Office of Science by Argonne National Laboratory under Contract No. DE-AC02-06CH11357. Additional crystallographic research was conducted at the Center for High Energy X-ray Sciences (CHEXS), Cornell High Energy Synchrotron Source (CHESS), which is supported by the NSF under award DMR-1829070. The MacCHESS resource is supported by NIGMS award 1-P30-GM124166-01A1 and NYSTAR. This work made use of the Cornell Center for Materials Research Shared Facilities, which are supported through the NSF MRSEC program (DMR-1719875). The funding sources were not involved in study design, data collection and interpretation, or the decision to submit the work for publication.

## Additional information

### Funding

| Funder | Grant reference number | Author |
|---|---|---|
| National Institutes of Health | R01AI142400 | Vincent T Lee |
| National Institutes of Health | R01GM123609 | Holger Sondermann |
| National Institutes of Health | R35GM136258 | J Christopher Fromme |

The funders had no role in study design, data collection and interpretation, or the decision to submit the work for publication.

### Author contributions

Justin D Lormand, Conceptualization, Data curation, Formal analysis, Investigation, Validation, Visualization, Writing - original draft, Writing - review and editing; Soo-Kyoung Kim, Conceptualization, Data curation, Formal analysis, Investigation, Validation, Visualization, Writing - review and editing; George A Walters-Marrah, Formal analysis, Investigation, Software, Validation, Writing - review and editing; Bryce A Brownfield, Data curation, Formal analysis, Investigation, Validation, Visualization, Writing - review and editing; J Christopher Fromme, Data curation, Formal analysis, Validation, Writing - review and editing; Wade C Winkler, Conceptualization, Funding acquisition, Validation, Writing - review and editing; Jonathan R Goodson, Conceptualization, Data curation, Formal analysis, Investigation, Software, Validation, Visualization, Writing - original draft, Writing - review and editing; Vincent T Lee, Holger Sondermann, Conceptualization, Data curation, Formal analysis, Funding acquisition, Investigation, Project administration, Supervision, Validation, Visualization, Writing - original draft, Writing - review and editing

### Author ORCIDs

Justin D Lormand http://orcid.org/0000-0002-7803-4716
J Christopher Fromme http://orcid.org/0000-0002-8837-0473
Vincent T Lee http://orcid.org/0000-0002-3593-0318
Holger Sondermann http://orcid.org/0000-0003-2211-6234

### Decision letter and Author response

Decision letter https://doi.org/10.7554/eLife.70146.sa1
Author response https://doi.org/10.7554/eLife.70146.sa2

## Additional files

### Supplementary files

- Supplementary file 1. X-ray data collection and refinement statistics.
- Supplementary file 2. Cryo-electron microscopy model validation statistics.
- Transparent reporting form

### Data availability

The atomic coordinates and structure factors have been deposited in the Protein Data Bank, https://www.rcsb.org/ (PDB ID codes 7MPL, 7MPM, 7MPN, 7MPO, 7MPP, 7MPQ, 7MPR, 7MPS, 7MPT, 7MPU, 7MQB/EMD-23941, 7MQD/EMD-23943, 7MQF/EMD-23945, 7MQH/EMD-23947, 7MQC/EMD-23942, 7MQE/EMD-23944, 7MQG/EMD-23946, 7MQI/EMD-23948).

The following dataset was generated:

| Author(s) | Year | Dataset title | Dataset URL | Database and Identifier |
|---|---|---|---|---|
| Lormand JD, Sondermann H | 2021 | Bartonella henselae NrnC bound to pGG | https://www.rcsb.org/structure/7MPL | RCSB Protein Data Bank, 7MPL |

| Author(s) | Year | Dataset title | Dataset URL | Database and Identifier |
|---|---|---|---|---|
| Lormand JD, Sondermann H | 2021 | Bartonella henselae NrnC bound to pAA | https://www.rcsb.org/structure/7MPM | RCSB Protein Data Bank, 7MPM |
| Lormand JD, Sondermann H | 2021 | Bartonella henselae NrnC bound to pGC | https://www.rcsb.org/structure/7MPN | RCSB Protein Data Bank, 7MPN |
| Lormand JD, Sondermann H | 2021 | Bartonella henselae NrnC bound to pAp | https://www.rcsb.org/structure/7MPO | RCSB Protein Data Bank, 7MPO |
| Lormand JD, Sondermann H | 2021 | Bartonella henselae NrnC cleaving pGG in the presence of Mg2+ | https://www.rcsb.org/structure/7MPP | RCSB Protein Data Bank, 7MPP |
| Lormand JD, Sondermann H | 2021 | Bartonella henselae NrnC cleaving pGG in the presence of Mn2+ | https://www.rcsb.org/structure/7MPQ | RCSB Protein Data Bank, 7MPQ |
| Lormand JD, Sondermann H | 2021 | Brucella melitensis NrnC | https://www.rcsb.org/structure/7MPR | RCSB Protein Data Bank, 7MPR |
| Lormand JD, Sondermann H | 2021 | Brucella melitensis NrnC with engaged loop | https://www.rcsb.org/structure/7MPS | RCSB Protein Data Bank, 7MPS |
| Lormand JD, Sondermann H | 2021 | Brucella melitensis NrnC with bound Mg2+ | https://www.rcsb.org/structure/7MPT | RCSB Protein Data Bank, 7MPT |
| Lormand JD, Sondermann H | 2021 | Brucella melitensis NrnC bound to pGG | https://www.rcsb.org/structure/7MPU | RCSB Protein Data Bank, 7MPU |
| Lormand JD, Brownfield B, Fromme JC, Sondermann H | 2021 | Bartonella henselae NrnC bound to pGG. D4 Symmetry | https://www.rcsb.org/structure/7MQB | RCSB Protein Data Bank, 7MQB/EMD-23941 |
| Lormand JD, Brownfield B, Fromme JC, Sondermann H | 2021 | Bartonella henselae NrnC complexed with pAGG. D4 symmetry | https://www.rcsb.org/structure/7MQD | RCSB Protein Data Bank, 7MQD/EMD-23943 |
| Lormand JD, Brownfield B, Fromme JC, Sondermann H | 2021 | Bartonella henselae NrnC complexed with pAAAGG. D4 symmetry | https://www.rcsb.org/structure/7MQF | RCSB Protein Data Bank, 7MQF/EMD-23945 |
| Lormand JD, Brownfield B, Fromme JC, Sondermann H | 2021 | Bartonella henselae NrnC complexed with pAAAGG in the presence of Ca2+. D4 Symmetry | https://www.rcsb.org/structure/7MQH | RCSB Protein Data Bank, 7MQH/EMD-23947 |
| Lormand JD, Brownfield B, Fromme JC, Sondermann H | 2021 | Bartonella henselae NrnC bound to pGG. C1 reconstruction | https://www.rcsb.org/structure/7MQC | RCSB Protein Data Bank, 7MQC/EMD-23942 |
| Lormand JD, Brownfield B, Fromme JC, Sondermann H | 2021 | Bartonella henselae NrnC complexed with pAGG. C1 reconstruction | https://www.rcsb.org/structure/7MQE | RCSB Protein Data Bank, 7MQE/EMD-23944 |
| Lormand JD, Brownfield B, Fromme JC, Sondermann H | 2021 | Bartonella henselae NrnC complexed with pAAAGG. C1 reconstruction | https://www.rcsb.org/structure/7MQG | RCSB Protein Data Bank, 7MQG/EMD-23946 |
| Lormand JD, Brownfield B, Fromme JC, Sondermann H | 2021 | Bartonella henselae NrnC complexed with pAAAGG in the presence of Ca2+. C1 reconstruction | https://www.rcsb.org/structure/7MQI | RCSB Protein Data Bank, 7MQI/EMD-23948 |

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
