## [Decision Letter]

**Decision letter after peer review:**

Thank you for submitting your article "Structural characterization of NrnC identifies unifying features of diribonucleases" for consideration by eLife. Your article has been reviewed by 3 peer reviewers, and the evaluation has been overseen by a Reviewing Editor and Gisela Storz as the Senior Editor. The following individuals involved in review of your submission have agreed to reveal their identity: Simon Dove (Reviewer #1); Arun Malhotra (Reviewer #2); Hanna S. Yuan (Reviewer #3).

Essential revisions:

The reviewers unanimously agree that this work is significant and appropriate for publication in *eLife* provided the authors revise the text in the following manner:

(1) Revision should include an additional experiment analyzing the effect of NrnC on the 3' overhang of a duplex RNA and duplex DNA as suggested by rev 2 and rev 3.

(2) Revision should include further discussion of possible roles for the central channel as suggested by rev 2.

*Reviewer #1:*

This interesting study by Lormand and colleagues focuses on nanoRNase C (NrnC), a 3'-5' exonuclease found in a variety of bacteria. This class of enzyme has been reported to degrade 2-7 nt RNA species during the final stages of RNA degradation. Whether NrnC exhibited any preference for a specific size of substrate was unclear. The manuscript provides X-ray crystallographic and cryoEM structural data that indicate NrnC is a dinuclease and illuminate the structural basis for this substrate specificity. The authors provide strong supporting biochemical and genetic evidence that NrnC acts preferentially as a dinuclease in vitro and in vivo. In addition, they show that the previously observed DNase activity of NrnC can only be observed under an unusual set of in vitro conditions that are unlikely to be physiologically relevant. This counters the idea that the function of NrnC, which is essential in some bacteria, is related to its ability to degrade DNA. The structural data in combination with the biochemical and genetic analyses of wild-type and mutant proteins (used to test structural predictions) support a model in which NrnC employs a narrow active site optimized for dinucleotide substrates, with a 'phosphate cap' coordinating the 5'-phophate of the substrate. The work is significant as it makes a strong case that NrnC acts by digesting dinucleotides rather than larger oligonucleotide species. It also reveals a striking parallel between NrnC and Orn, another previously ascribed member of the nanoRNase family that this same group elegantly demonstrated was a dedicated dinuclease. Although Orn and NrnC both belong to the DnaQ family of ribonucleases, phylogenetic analyses reveal they are evolutionarily distinct with their substrate specificities arising independently through convergent evolution. Taken together with the available genetic evidence that NrnC and Orn are essential in many bacteria, the work bolsters the idea that dinucleotide degradation is a key final step in the degradation of RNA species in bacteria.

*Reviewer #2:*

This manuscript looks at the structure and function of NrnC, one of the three nanoRNases that degrade small RNA fragments in the final step of RNA degradation. The other two, Orn (oligoribonuclease) and NrnA/B (NanoRNase A/B) belong to two different protein families, DEDDh and DHH1-DHHA1 nuclease families. NrnC shares the same DnaQ fold and DEDD motif as oligoribonuclease. However, it forms an octameric assembly rather than the dimeric arrangement seen for Orn (Yuan et al., 2018). NrnC's specific activity also has not been well characterized.

The authors of this manuscript recently showed that Orn is diribonuclease and a key enzyme for the terminal step of RNA degradation (Kim et al., *eLife* 2019;8:e46313). In this manuscript they show that NrnC also works as a diribonuclease and shares the same substrate preference as Orn. The authors solve X-ray structures of NrnC from B. henselae and B. melitensis in multiple configurations and crystals forms to understand how NrnC binds dinucleotides. Features seen in Orn – a leucine wedge and a phosphate cap are also present in NrnA, but with a different sequence motif for the P-Cap. To look at binding of larger RNA substrates, the authors use cryo-EM but this does not add much as only the 3' dinucleotide regions are well ordered and visible. Indeed, one weakness of the manuscript is the large multitude of structures, all essentially providing the same insights. Much of this is presented in the supplementary figures, but it detracts from the main points the authors are presenting.

Since the longer RNA substrates could not be visualized, one important unanswered question is the role of the octameric architecture and the narrow central channel. Many proteolytic and nucleolytic machines use a well defined central channel to carefully select and feed substrates to potent but non-selective degradative active sites that are sequestered within their ologomeric assembly. If NrnC predominately works on dinucleotides, why does it need a central channel to direct such small substrates to the eight active sites. Is this a mechanism to "throttle" the enzyme? Some discussion of possible roles for the channel would be useful.

Results from the structural studies are corroborated by functional studies (in-vitro activity assays and substate binding studies, as well as in-vivo complementation assays). These show a strong preference of NrnC for dinucleotide substrates. Were 3' sequences other then GG also tested?

A major contribution here is the phylogenetic analysis that suggests that NrnC may have developed its preference for dinucleotides via convergent evolution independent of Orn, and much later than the split between the DEDDh Orn and DEDDy NrnC/RNase D protein families. Figure 8 nicely summarizes the differences as well as common motifs between Orn and NrnC.

A few other points to address:

– Line 144 and Figure 1 Supplement 2 – Did the authors mean to compare NrnC to the Rrp6 Exo domain rather than the Klenow fragment? PDB 4oo1 and Wasmuth et al., 2014 describe the structure of Rrp6

– Figure 4, supplement figure 5-8 – Is there any significance to the use of different colors for the SKQQQS loop region between these figures? In general, does the SKQQS loop act as an "auto-inhibitory" loop that has to be displaced by the incoming substrate?

– Lines 324-330 – Was DNA substrate with 3' overhang tested? If the enzyme works 3'->5', a 3' overhang may be a more appropriate substrate.

*Reviewer #3:*

This manuscript reports the structures of NrnC from two bacteria showing how this enzyme specifically binds and degrades only nanoRNA containing two ribonucleotides. Overall, this manuscript provides the structural basis for the specificity of NrnC with a restricted active site for binding and cleaving di-ribonucleotides. However, this study did not address the key question of why NrnC has an octameric assembly that is very different from the dimeric assembly of ORN, which shares a similar DEDD-type structure and diribonuclease function to NrnC.

The major concern is that this manuscript did not address the key question of why NrnC is evolved into an octamer but not a dimer. This octameric assembly is interesting as it contains a central channel that is ideal for binding a double helix of RNA or DNA (see the model in the reference of Yuan et al., 2018). The authors did not test if NrnC cleaves the 3' overhang of a duplex RNA, and if NrnC cleaves the 3' overhang of a duplex DNA and possibly participates in RNA maturation or DNA repair. The octameric assembly is conserved for NrnC in different bacterial strains, suggesting that it has a reason for the enzyme to evolve as an octamer but not a homo-dimer.

It is shown in Figure 5—figure supplement 2 that NrnC cleaves long dsDNA only in the presence of Mn^2+^. A short DNA was not used for the degradation experiments. For example, it is not clear if NrnC can cleave nanoDNA of two to five nucleotides. These short nanoDNA could be generated as byproducts during DNA repair. Based on the crystal structures of NrnC, it seems that NrnC should be able to degrade a di-nucleotide DNA.

---

## [Author Response]

Essential revisions:The reviewers unanimously agree that this work is significant and appropriate for publication in eLife provided the authors revise the text in the following manner:(1) Revision should include an additional experiment analyzing the effect of NrnC on the 3' overhang of a duplex RNA and duplex DNA as suggested by rev 2 and rev 3.

We have conducted the suggested experiments (Figure 5—figure supplement 3 of the revised manuscript). As you will see, under conditions where we observe rapid turnover of dinucleotides, we do not observe appreciable degradation of double-stranded substrates. At this point, and as stated in our manuscript, we cannot rule out specific conditions under which NrnC may act on double-stranded DNA, but our collective data suggest dinucleotides as the highly preferred substrate of NrnC.

(2) Revision should include further discussion of possible roles for the central channel as suggested by rev 2.

We extended the discussion regarding the relevance of the channel that NrnC octamers present. We speculate that the channel may provide a nano-compartment for the degradation of dinucleotides, with positively-charged residues lining the channel attracting the substrate. Furthermore, the constrictions a small channel would provide could also sterically limit accidental degradation of other, larger nucleotide substrates.

Reviewer #2:This manuscript looks at the structure and function of NrnC, one of the three nanoRNases that degrade small RNA fragments in the final step of RNA degradation. The other two, Orn (oligoribonuclease) and NrnA/B (NanoRNase A/B) belong to two different protein families, DEDDh and DHH1-DHHA1 nuclease families. NrnC shares the same DnaQ fold and DEDD motif as oligoribonuclease. However, it forms an octameric assembly rather than the dimeric arrangement seen for Orn (Yuan et al., 2018). NrnC's specific activity also has not been well characterized.The authors of this manuscript recently showed that Orn is diribonuclease and a key enzyme for the terminal step of RNA degradation (Kim et al., eLife 2019;8:e46313). In this manuscript they show that NrnC also works as a diribonuclease and shares the same substrate preference as Orn. The authors solve X-ray structures of NrnC from B. henselae and B. melitensis in multiple configurations and crystals forms to understand how NrnC binds dinucleotides. Features seen in Orn – a leucine wedge and a phosphate cap are also present in NrnA, but with a different sequence motif for the P-Cap. To look at binding of larger RNA substrates, the authors use cryo-EM but this does not add much as only the 3' dinucleotide regions are well ordered and visible. Indeed, one weakness of the manuscript is the large multitude of structures, all essentially providing the same insights. Much of this is presented in the supplementary figures, but it detracts from the main points the authors are presenting.

We acknowledge the large volume of our study and how this could be perceived as distracting by some. However, each structural analysis by crystallography and EM provides an important piece in elucidating NrnC’s substrate specificity (e.g. by demonstrating the existence of a flexible activation loop; providing confirmation that a catalytically active state was crystallized; and establishing that the active site appears optimized for dinucleotides).

Since the longer RNA substrates could not be visualized, one important unanswered question is the role of the octameric architecture and the narrow central channel. Many proteolytic and nucleolytic machines use a well defined central channel to carefully select and feed substrates to potent but non-selective degradative active sites that are sequestered within their ologomeric assembly. If NrnC predominately works on dinucleotides, why does it need a central channel to direct such small substrates to the eight active sites. Is this a mechanism to "throttle" the enzyme? Some discussion of possible roles for the channel would be useful.

See above. We added to the discussion regarding the role of the octameric channel formed by NrnC. One possibility is that the channel provides a nano-compartment in which dinucleotides are efficiently sequestered and subsequently degraded. The narrow opening may sterically hinder the access of longer or double-stranded nucleotides.

Results from the structural studies are corroborated by functional studies (in-vitro activity assays and substate binding studies, as well as in-vivo complementation assays). These show a strong preference of NrnC for dinucleotide substrates. Were 3' sequences other then GG also tested?

We originally only tested activity against nucleotides with 3’ GG. However, we presented structures of the enzyme bound to pAA and pCC. The binding poses are similar to the one of pGG at NrnC, which indicates less of an impact of the dinucleotide sequence. An additional experiment in the revised manuscript shows that pAA is degraded with the same kinetics as pGG (Figure 5—figure supplements 3E), supporting this interpretation further.

A major contribution here is the phylogenetic analysis that suggests that NrnC may have developed its preference for dinucleotides via convergent evolution independent of Orn, and much later than the split between the DEDDh Orn and DEDDy NrnC/RNase D protein families. Figure 8 nicely summarizes the differences as well as common motifs between Orn and NrnC.A few other points to address:– Line 144 and Figure 1 Supplement 2 – Did the authors mean to compare NrnC to the Rrp6 Exo domain rather than the Klenow fragment? PDB 4oo1 and Wasmuth et al., 2014 describe the structure of Rrp6

Yes, we used the Rrp6 exonuclease domain of PDB 4oo1 for the analysis. We corrected this mistake in the figure, figure legend and text of the revised manuscript. Thank you for catching this mistake.

– Figure 4, supplement figure 5-8 – Is there any significance to the use of different colors for the SKQQQS loop region between these figures? In general, does the SKQQS loop act as an "auto-inhibitory" loop that has to be displaced by the incoming substrate?

The color coding refers to individual structure determination, different in the length of the RNA substrate used. We added a clarifying statement to Figure Legend 4, where the coloring was introduced first.

Based on our analysis, we would view the SKQQQS loop more as an activation loop that needs to swing into the active site, adopting an ordered conformation that is accompanied with positioning the catalytic tyrosine residue of the DEDDy motif in a catalytically competent position (also illustrated in Figure 3—figure supplement 1). At the same time, it likely would have to move out of the way to allow substrate binding. We included this discussion in the revised manuscript.

– Lines 324-330 – Was DNA substrate with 3' overhang tested? If the enzyme works 3'->5', a 3' overhang may be a more appropriate substrate.

KpnI-cleaved DNA substrate in the original manuscript created a 3’-overhang. No processive activity was observed with this substrate. In the revised manuscript, we extended this analysis in response to points also raised by Reviewer #3 (see below and Figure 5—figure supplement 3 in the revised manuscript).

Reviewer #3:This manuscript reports the structures of NrnC from two bacteria showing how this enzyme specifically binds and degrades only nanoRNA containing two ribonucleotides. Overall, this manuscript provides the structural basis for the specificity of NrnC with a restricted active site for binding and cleaving di-ribonucleotides. However, this study did not address the key question of why NrnC has an octameric assembly that is very different from the dimeric assembly of ORN, which shares a similar DEDD-type structure and diribonuclease function to NrnC.

NrnC certainly forms a peculiar octameric assembly, quite different from the dimers formed by Orn and Rexo2. In the revised manuscript, we discuss possible advantages for an octameric assembly (see above). Besides serving as a reaction chamber for efficient dinucleotide processing, an octameric assembly may increase the stability of the enzyme. At the same time, one may argue that it is also not clear why Orn and Rexo2 form dimers as biologically active units as there is no apparent sign of cooperativity between the two active sites of a dimer.

The major concern is that this manuscript did not address the key question of why NrnC is evolved into an octamer but not a dimer. This octameric assembly is interesting as it contains a central channel that is ideal for binding a double helix of RNA or DNA (see the model in the reference of Yuan et al., 2018). The authors did not test if NrnC cleaves the 3' overhang of a duplex RNA, and if NrnC cleaves the 3' overhang of a duplex DNA and possibly participates in RNA maturation or DNA repair. The octameric assembly is conserved for NrnC in different bacterial strains, suggesting that it has a reason for the enzyme to evolve as an octamer but not a homo-dimer.

We included the requested experiments in the revised manuscript (Figure 5—figure supplement 3). These experiments clearly establish that the NrnC orthologs tested here do not degrade double-stranded DNA or RNA, neither blunt nor with a 3’ overhang. Under identical conditions, dinucleotides are degraded efficiently. Longer, single-stranded nucleotides are degraded albeit at much reduced efficiency. Together, these experiments further support our previous results and their interpretations.

It is shown in Figure 5—figure supplement 2 that NrnC cleaves long dsDNA only in the presence of Mn^2+^. A short DNA was not used for the degradation experiments. For example, it is not clear if NrnC can cleave nanoDNA of two to five nucleotides. These short nanoDNA could be generated as byproducts during DNA repair. Based on the crystal structures of NrnC, it seems that NrnC should be able to degrade a di-nucleotide DNA.

We show in the revised manuscript that NrnC cleaves nanoDNA two nucleotides in length at the same rate as diribonucleotides of the same sequence. As observed with RNA, longer, single-stranded DNA, such as trinucleotide, are appreciable poorer substrates (Figure 5—figure supplement 3). Considering this new result, we changed descriptions, where appropriate, regarding NrnC’s activity throughout the amended manuscript to reflect this broader activity (i.e., dinuclease instead of diribonuclease activity).